# Generalization bound of globally optimal non-convex neural network training: Transportation map estimation by infinite dimensional Langevin dynamics

**Taiji Suzuki**
The University of Tokyo, Tokyo, Japan
RIKEN Center for Advanced Intelligence Project, Tokyo, Japan
`taiji@mist.i.u-tokyo.ac.jp`

## Abstract

We introduce a new theoretical framework to analyze deep learning optimization with connection to its generalization error. Existing frameworks such as mean field theory and neural tangent kernel theory for neural network optimization analysis typically require taking limit of infinite width of the network to show its global convergence. This potentially makes it difficult to directly deal with finite width network; especially in the neural tangent kernel regime, we cannot reveal favorable properties of neural networks beyond kernel methods. To realize more natural analysis, we consider a completely different approach in which we formulate the parameter training as a transportation map estimation and show its global convergence via the theory of the *infinite dimensional Langevin dynamics*. This enables us to analyze narrow and wide networks in a unifying manner. Moreover, we give generalization gap and excess risk bounds for the solution obtained by the dynamics. The excess risk bound achieves the so-called fast learning rate. In particular, we show an exponential convergence for a classification problem and a minimax optimal rate for a regression problem.

## 1 Introduction

Despite the extensive empirical success of deep learning, there are several missing issues in theoretical understanding of its optimization and generalizations. Even though there are several theoretical analyses on its generalization error and representation ability [46, 8, 2, 67, 56], they are not necessarily well connected with an optimization procedure. The biggest difficulty in neural network optimization lies in its non-convexity. Recently, this difficulty of non-convexity is partly resolved by considering infinite width limit of networks as performed in *mean field theory* [58, 40] and *Neural Tangent Kernel* (NTK) [32, 22]. These analyses deal with different scaling of parameters for taking the limit of the width, but they share a similar spirit that an appropriate gradient descent direction can be found in an over-parameterized setting until convergence.

The mean field analysis formulates the neural network training as a gradient flow in the space of probability measures over the weights. The gradient flow corresponding to a deterministic dynamics of the weights can be analyzed as an interacting particle system [47, 18, 53, 54]. On the other hand, a stochastic dynamics of an interacting particle system can be formulated as McKean–Vlasov dynamics, and convergence to the global optimal is ensured by the ergodicity of this dynamics [40, 41]. Intuitively, inducing stochastic noise makes the solution easier to get out of local optimal and facilitates convergence to the global optimal.

The second regime, NTK, deals with larger scaling than the mean field regime, and the gradient descent dynamics is approximated by that in the tangent space at the initial solution [32, 23, 1, 22, 3].

That is, in the wide limit of the neural network, the gradient descent can be seen as that in an reproducing kernel Hilbert space (RKHS) corresponding to the neural tangent kernel, which resolves the difficulty of non-convexity. Actually, it is shown that the gradient descent converges to the zero error solution exponentially fast for a sufficiently large width network [23, 1, 22]. In addition to the optimization, its generalization error has been also extensively studied in the NTK regime [23, 1, 22, 76, 16, 17, 79, 50, 48, 34]. On the other hand, [29] pointed out that non-convexity of a deep neural network model is essential to show superiority of deep learning over linear estimators such as kernel methods as in the analysis of [65, 30, 66]. Therefore, the NTK regime would not be appropriate to show superiority of deep learning over other methods such as kernel methods.

The above mentioned researches opened up new directions for analyzing deep learning optimization. However, all of them require that the width should diverge as the sample size goes up to show the global convergence and obtain generalization error bounds. On the other hand, a convergence guarantee for "fixed width" training is still difficult and we have not obtained a satisfactory result that can bridge both of under-parameterized and over-parameterized settings in a *unifying manner*. One way to tackle non-convexity in a finite width situation would be stochastic gradient Langevin dynamics (SGLD) [77, 51, 24]. This would be useful to show the global convergence for the non-convex optimization in deep leaning. However, the convergence rate depends exponentially to the dimensionality, which is not realistic to analyzing neural network training that typically requires huge parameter size.

**Our contribution:** In this paper, we resolve these difficulties such as (i) diverging width against sample size and (ii) curse of dimensionality for analyzing Langevin dynamics in neural network training by formulating the neural network training as a *transport map* estimation problem of the parameters. By doing so, we can deal with finite width and infinite width in a unifying manner. We also give a generalization error bound for the solution obtained by our optimization formulation and further show that it achieves *fast learning rate* in a well-specified setting. The preferable generalization error heavily relies on similarity between a *nonparametric Bayesian Gaussian process estimator* and the Langevin dynamics. More details are summarized as follows:

- **(formulation)** We formulate neural network training as a transportation map learning of weights (parameters) and solve this problem by infinite dimensional gradient Langevin dynamics in RKHS [20, 45]. This formulation has a wide range of applications including two layer neural network, ResNet, Wasserstein optimal transportation map estimation and so on.

- **(optimization)** Based on this formulation, we show its global convergence for finite width and infinite width in a unifying manner. We give its size independent convergence rate.

- **(generalization)** We derive the generalization error bound of the estimator obtained by our optimization framework. We also derive the fast learning rate in a student-teacher setup. Especially, we show exponential convergence for classification.

## 2 Problem setting and model: Training parameter transportation map

In this section, we give the problem setting and notations that will be used in the theoretical analysis. Basically, we consider the standard supervised leaning where data consists of input-output pairs $z = (x, y)$ where $x \in \mathbb{R}^d$ is an input and $y \in \mathbb{R}$ is an output (or label). We may also consider a unsupervised learning setting, but just for the presentation simplicity, we consider a supervised learning. Suppose that we are given $n$ i.i.d. observations $D_n = (x_i, y_i)_{i=1}^n$ distributed from a probability distribution $P$, the marginal distributions of which with respect to $x$ and $y$ are denoted by $P_X$ and $P_Y$ respectively. We denote $\mathcal{X} = \mathrm{supp}(P_X)$. To measure the performance of a trained function $f$, we use a loss function $\ell : \mathbb{R} \times \mathbb{R} \to \mathbb{R}$ $((y, f) \mapsto \ell(y, f))$ and define the expected risk and the empirical risk as $\mathcal{L}(f) := \mathrm{E}_{Y,X}[\ell(Y, f(X))]$ and $\widehat{\mathcal{L}}(f) := \frac{1}{n} \sum_{i=1}^n \ell(y_i, f(x_i))$ respectively. As in the standard deep learning, we optimize the training risk $\widehat{\mathcal{L}}$. Our theoretical interest is to bound the following errors for an estimator $\widehat{f}$:

$$\text{Excess risk: } \mathcal{L}(\widehat{f}) - \inf_{f:\text{measurable}} \mathcal{L}(f), \quad \text{Generalization gap: } \mathcal{L}(\widehat{f}) - \widehat{\mathcal{L}}(\widehat{f}).$$

In a typical situation, the generalization gap is bounded as $O(1/\sqrt{n})$ via VC-theory type analysis [43], for example. On the other hand, the excess risk can be faster than $O(1/\sqrt{n})$, which is known as a *fast learning rate* [42, 5, 35, 27]. The population $L_2$-norm with respect to $P$ is denoted by

$\|f\|_{L_2} := \sqrt{\mathbb{E}_{Z \sim P}[f(Z)^2]}$ and the sup-norm on the domain of the input distribution $P_X$ is denoted by $\|f\|_\infty := \sup_{x \in \text{supp}(P_X)} |f(x)|$.

## 2.1 Introductory setting: mean field training of two layer neural network

Here, we explain the motivation of our theoretical framework by introducing mean field analysis of two layer neural networks. Let us consider the following two layer neural network model:

$$f_\Theta(x) = \frac{1}{M} \sum_{m=1}^M a_m \sigma(w_m^\top x). \tag{1}$$

where $\sigma : \mathbb{R} \to \mathbb{R}$ is a smooth activation function, $(a_m)_{m=1}^M \subset \mathbb{R}$ is the set of weights in the second layer which we assume is fixed for simplicity, and $\Theta = (w_m)_{m=1}^M \subset \mathbb{R}^d$ is the set of weights in the first layer. We aim to minimize the following regularized empirical risk with respect to $\Theta$ and analyze the dynamics of gradient descent updates:

$$\min_\Theta \ \widehat{\mathcal{L}}(f_\Theta) + \frac{\lambda}{2M} \sum_{m=1}^M \|w_m\|^2.$$

The stochastic gradient descent (SGD) update for optimizing $\widehat{\mathcal{L}}(f_\Theta)$ with respect to $\Theta$ is reduced to

$$w_m^{(t+1)} = w_m^{(t)} - \eta\left(\frac{\lambda}{M} w_m^{(t)} + \nabla_{w_m} \widehat{\mathcal{L}}(f_{\Theta^{(t)}})\right) + \sqrt{2\eta/\beta} \epsilon_t^{(m)}, \tag{2}$$

where $\nabla_{w_m} \widehat{\mathcal{L}}(f_{\Theta^{(t)}}) = \frac{a_m}{M} \frac{1}{n} \sum_{i=1}^n x_i \sigma'(w_m^{(t)\top} x_i) \ell'(y_i, f_{\Theta^{(t)}}(x_i))$ and $\epsilon_t^{(m)}$ is an i.i.d. Gaussian noise mimicking the deviation of the stochastic gradient. Here, $\eta > 0$ is a step size and $\beta > 0$ is an inverse temperature parameter. This could be time discretized version of the following continuous time stochastic differential equation (SDE):

$$\mathrm{d}w_m(t) = -\left(\frac{\lambda}{M} w_m(t) + \nabla_{w_m(t)} \widehat{\mathcal{L}}(f_{\Theta^{(t)}})\right)\mathrm{d}t + \sqrt{2\eta/\beta}\,\mathrm{d}B_t^{(m)},$$

where $(B_t^{(m)})_t$ is a $d$-dimensional Brownian motion. In the mean field analysis, this optimization process is casted to an optimization of probability distribution over the parameters [40, 41, 47, 18] based on the following integral representation of neural networks:

$$f_\rho(x) := \int_{\mathbb{R}^d} a\sigma(w^\top x)\mathrm{d}\rho(w), \tag{3}$$

where $\rho$ is a Borel probability measure defined on the parameter space $\mathbb{R}^d$ and the parameter in the second layer is fixed to a constant $a \in \mathbb{R}$ just for presentation simplicity. The time evolution of the distribution $\rho$ is deduced from the optimization dynamics with respect to each "particle" given by

$$\mathrm{d}W(t) = -\left(\lambda W(t) + a\frac{1}{n} \sum_{i=1}^n x_i \sigma'(W(t)^\top x_i) \ell'(y_i, f_{\rho_t}(x_i))\right)\mathrm{d}t + \sqrt{\beta^{-1}}\mathrm{d}B_t,$$

where $\rho_t$ is the probability law of $W(t) \in \mathbb{R}^d$ with an initial distribution $W(0) \sim \rho_0$, which is one of the *McKean-Vlasov* processes. We can see that this equation is space-time continuous limit of the update Eq. (2). Importantly, $\rho_t$ admits a density function $\pi_t$ obeying the so-called continuity equation [40, 41]. The usual finite width network is regarded as a finite sum approximation of the integral representation (Eq. (3)). As a consequence, the convergence analysis needs to take limit of infinite width to approximate the absolutely continuous distribution $\rho_t$. Hence, a finite width dynamics is outside the scope of mean field analysis. This is due to the fact that an independent noise is injected to each particle regardless its location; the diffusion $B_t$ is independently and identically applied to each realized path $\{W(t) \mid t \geq 0\}$ (interaction between particles is induced only through gradient). However, in a real neural network training, the noise induced by stochastic gradient has high correlation between each node. Thus, we need a different approach.

**Lift of McKean-Vlasov process** Our core idea is to "lift" the stochastic process $W(t)$ as a process of a function with the initial value $W(0)$. For each $W(0) = w_0$, the particle's location at time $t$ is determined by $W(t) = W(t, w_0)$. This means that the process generates a function $w_0 \mapsto W(t, w_0)$ with respect to the initial solution $w_0$. By considering the stochastic process of this function itself directly, the dynamics is transformed to an *infinite dimensional stochastic differential equation*, which has been studied especially in the stochastic partial differential equation [20]. In other words,

we try to estimate a map from the initial parameters to the solution at time $t$ instead of analyzing each particle's behavior.

From this perspective, we can directly regularize the smoothness of the trajectory, especially, we can incorporate a smoothed noise of the dynamics by utilizing a spatially correlated Gaussian process in the space of functions on parameters. Let $W_t(w) = W(t, w)$ and we regard $W_t$ as a member of $L_2(\rho_0)$ space. Then, $f_{\rho_t}$ can be rewritten by

$$f_{W_t}(x) := \int_{\mathbb{R}^d} a\sigma(W_t(w)^\top x)\mathrm{d}\rho_0(w) = \int_{\mathbb{R}^d} a\sigma(w^\top x)\mathrm{d}W_t\sharp\rho_0(w), \qquad (4)$$

where $W_t\sharp\rho_0$ is the pushforward of the measure $\rho_0$ by the map $W_t$, i.e., $f\sharp\mu(B) := \mu \circ f^{-1}(B) = \mu(f^{-1}(B))$ for a Borel measurable map $f : \mathbb{R}^d \to \mathbb{R}^d$, a Borel measure $\mu$, and a Borel set $B \subset \mathbb{R}^d$. By using this notation, the stochastic process we consider can be written as

$$\mathrm{d}W_t = -(AW_t + \nabla_W\widehat{\mathcal{L}}(f_{W_t}))\mathrm{d}t + \sqrt{2\beta^{-1}}\mathrm{d}\xi_t, \qquad (5)$$

where $A : L_2(\rho_0) \to L_2(\rho_0)$ is an unbounded linear operator corresponding to a regularization (which will be explained later in more details), $\nabla_W\widehat{\mathcal{L}}(f_W)$ is the Frechet derivative of $\widehat{\mathcal{L}}(f_W)$ with respect to $W$ in the space of $L_2(\rho_0)$, in our setting, which is given by $\nabla_W\widehat{\mathcal{L}}(f_W)(w) = a\frac{1}{n}\sum_{i=1}^n x_i\sigma'(W(w)^\top x_i)\ell'(y_i, f_W(x_i))$. $(\xi_t)_t$ is a *cylindric Brownian motion* in $L_2(\rho_0)$ [20], which is an infinite dimensional Brownian motion and will be defined rigorously later on. In practical deep learning, the regularization term $AW_t$ is induced by several mechanism such as weight decay [37], dropout [60, 74], batch-normalization [31]. As a result, the regularization term $AW_t$ introduces spatial correlation between particles unlike the McKean-Vlasov process.

Then, training two layer neural networks is formulated as optimizing the map $W : w \in \mathbb{R}^d \mapsto W(w) \in \mathbb{R}^d$ with the initial condition $W_0 = \mathbb{I}$ (identity map). This dynamics is well analyzed and guaranteed to converge to at least a stationary distribution (a.k.a., invariant measure) under mild assumptions [19, 39, 59, 33, 57, 28] which is useful to show convergence to a (near) global optimal.

**Remark 1.** *We would like to emphasize that our formulation admits a finite width neural network training by setting the initial distribution $\rho_0$ as a discrete distribution $\rho_0 = \frac{1}{M}\sum_{m=1}^M \delta_{w_m}$ for a Dirac measure $\delta_{w_m}$ which has probability 1 on a point $w_m$. In this situation, optimizing the map $W_t$ corresponds to optimizing the finite width model* (1) *because $\rho_t = W_t\sharp\rho_0 = \frac{1}{M}\sum_{m=1}^M \delta_{W_t(w_m)}$ which is still a discrete distribution throughout entire $t \in \mathbb{R}_+$. This is remarkably different from both mean field analysis and NTK analysis that essentially take infinite width limits: mean field analysis in [40, 41] requires $M = \Omega(e^T)$ for a time horizon $T$ and NTK requires $M = \Omega(\mathrm{poly}(n))$ [79].*

**General formulation of our optimization problem** Here, we describe mathematical details of optimizing the transportation map in a more general setting and give a practical algorithm of the corresponding GLD. We assume that the map $W_t(\cdot)$ is included in a separable Hilbert space $\mathcal{H}$ with norm $\|\cdot\|_{\mathcal{H}}$ and an inner product $\langle\cdot,\cdot\rangle_{\mathcal{H}}$ (in the previous section, $\mathcal{H} = L_2(\rho_0)$). The Hilbert space $\mathcal{H}$ consists of functions whose domain is a set $\mathcal{W}$ and whose range is $\widetilde{\mathcal{W}}$ (in the previous example, $\mathcal{W} = \mathbb{R}^d$ amd $\widetilde{\mathcal{W}} = \mathbb{R}^d$). Since a function $w \in \mathcal{H}$ has no smoothness condition in typical settings, we consider a more "regulated" subspace of $\mathcal{H}$. Such a subspace is denoted by $\mathcal{H}_K$ and given by $\mathcal{H}_K := \left\{\sum_{k=0}^\infty \alpha_k e_k \mid \sum_{k=0}^\infty \alpha_k^2/\mu_k < \infty\right\}$, where $(e_k)_{k=0}^\infty$ is an orthonormal basis of $\mathcal{H}$ and $(\mu_k)_{k=0}^\infty$ is a non-increasing non-negative sequence. We equip an inner product $\langle\cdot,\cdot\rangle_{\mathcal{H}_K}$ to the space $\mathcal{H}_K$ defined by $\langle f, g\rangle_{\mathcal{H}_K} = \sum_{k=0}^\infty \alpha_k\beta_k/\mu_k$ for $f = \sum_{k=0}^\infty \alpha_k e_k \in \mathcal{H}_K$ and $g = \sum_{k=0}^\infty \beta_k e_k \in \mathcal{H}_K$. Correspondingly, the norm $\|\cdot\|_{\mathcal{H}_K}$ is defined from the inner product. When $\mathcal{H} = L_2(\rho_0)$, $\mathcal{H}_K$ becomes a *reproducing kernel Hilbert space* (RKHS) corresponding to a kernel function $K(x, y) = \sum_{k=0}^\infty \mu_k e_k(x)e_k(y)$ where $x, y \in \mathbb{R}^d$ under an appropriate convergence condition. That is, we have the reproducing property $\langle K(x, \cdot), W\rangle_{\mathcal{H}_K} = W(x)$ for each $W \in \mathcal{H}_K$. Based on the norm $\|\cdot\|_{\mathcal{H}_K}$, we define an unbounded linear operator $A : \mathcal{H} \to \mathcal{H}$ as $Af = \lambda\sum_{k=0}^\infty \frac{\alpha_k}{\mu_k}e_k$, for $f = \sum_{k=0}^\infty \alpha_k e_k \in \mathcal{H}$. We note that $Af = \frac{\lambda}{2}\nabla_f\|f\|_{\mathcal{H}_K}^2$ which is a Frechet derivative of $\lambda\|\cdot\|_{\mathcal{H}_K}^2$ in $\mathcal{H}$ (which is the derivative of the RKHS norm, if $\mathcal{H}_K$ is an RKHS). We assume that for each $W \in \mathcal{H}$, there exits a function $f_W : \mathbb{R}^d \to \mathbb{R}$ as in Eq. (4), and we basically aim to minimize the regularized empirical risk

$$\widehat{\mathcal{L}}(f_W) + \frac{\lambda}{2}\|W\|_{\mathcal{H}_K}^2.$$

By abuse of notation, we denote by $\widehat{\mathcal{L}}(W)$ indicating $\widehat{\mathcal{L}}(f_W)$. To execute this non-convex optimization, we use the GLD in the infinite dimensional Hilbert space $\mathcal{H}$ as introduced in Eq. (5). Here, $(\xi_t)_{t \geq 0}$ in Eq. (5) is the cylindrical Brownian motion defined as $\xi_t = \sum_{k \geq 0} B_t^{(k)} e_k$ where $(B_t^{(k)})_{t \geq 0}$ is a real valued standard Brownian motion and they are independently identical for $k = 0, 1, 2, \ldots$[1]. Since this is defined on a continuous time domain, we introduce a discrete time *implicit Euler scheme* for practical implementation:

$$W_{k+1} = W_k - \eta(AW_{k+1} + \nabla_W \widehat{\mathcal{L}}(W_k)) + \sqrt{\tfrac{2\eta}{\beta}}\epsilon_k \Leftrightarrow W_{k+1} = S_\eta\left(W_k - \eta\nabla_W \widehat{\mathcal{L}}(W_k) + \sqrt{\tfrac{2\eta}{\beta}}\epsilon_k\right), \quad (6)$$

where $\eta > 0$ is the step size and $S_\eta = (\mathbb{I} + \eta A)^{-1}$. We can see that the "regularization effect" $AW$ induces the spacial smoothness of the noise of the gradient. It is known [14] that under some assumption (Assumption 1 below is sufficient), the process (5) has a unique invariant measure $\pi_\infty$ given by

$$\frac{\mathrm{d}\pi_\infty}{\mathrm{d}\nu_\beta}(W) \propto \exp(-\beta\widehat{\mathcal{L}}(W)),$$

where $\nu_\beta$ is the Gaussian measure in $\mathcal{H}$ with mean 0 and covariance $(\beta A)^{-1}$ (see Da Prato & Zabczyk [20] for the rigorous definition of the Gaussian measure on a Hilbert space and related topics about existence of invariant measure). In a special situation where $\beta = n$, $\lambda = 1/n$ and $\beta\widehat{\mathcal{L}}(W)$ is a log-likelihood function of some model, this invariant measure is nothing but the *Bayes posterior distribution* for a Gaussian process prior corresponding to the RKHS $\mathcal{H}_K$. Remarkably, this formulation can be applied to several problems other than training two layer neural networks:

- Ordinary nonparametric regression model: $\mathcal{W} = \mathbb{R}^d$, $\widetilde{\mathcal{W}} = \mathbb{R}$ and $f_W(x) = W(x)$.
- Two layer neural networks (continuous topology): $\mathcal{W} = \widetilde{\mathcal{W}} = \mathbb{R}^d$ and $f_W = \int_{\mathbb{R}^d} a(w)\sigma(W(w)^\top x)\mathrm{d}\rho_0(w)$.
- Two layer neural networks (discrete topology): $\mathcal{W} = \{1, 2, 3, \ldots\}$, $\widetilde{\mathcal{W}} = \mathbb{R}^d$ and $f_W = \sum_{m=1}^\infty a_m\sigma(W(m)^\top x)$.
- Two layer neural networks (discrete topology): $\mathcal{W} = \{1, 2, 3, \ldots\}$, $\widetilde{\mathcal{W}} = \mathbb{R}^d$ and $f_W = \sum_{m=1}^\infty a_m\sigma(W(m)^\top x)$.
- Deep neural networks (continuous topology): $\mathcal{W} = \mathbb{R}^d \times \{1, \ldots, L\}$, $\widetilde{\mathcal{W}} = \mathbb{R}^d$ and

$$f_W(x) = u^\top \left(\int_{\mathbb{R}^d} a_{w,L}\sigma(W(w,L)^\top \cdot)\mathrm{d}\rho_0(w)\right) \circ \cdots \circ \left(\int_{\mathbb{R}^d} a_{w,1}\sigma(W(w,1)^\top x)\mathrm{d}\rho_0(w)\right),$$

where $u \in \mathbb{R}^d$ and $a_{w,\ell} \in \mathbb{R}^d$ for $w \in \mathbb{R}^d$ and $\ell \in \{1, \ldots, L\}$.

- ResNet: $\mathcal{W} = \mathbb{R}^d \times \{1, \ldots, T\}$, $\widetilde{\mathcal{W}} = \mathbb{R}^d$ and

$$f_W(x) = u^\top \left(\mathbb{I} + \int_{\mathbb{R}^d} a_{w,T}\sigma(W(w,T)^\top \cdot)\mathrm{d}\rho_0(w)\right) \circ \cdots \circ \left(\mathbb{I} + \int_{\mathbb{R}^d} a_{w,1}\sigma(W(w,1)^\top x)\mathrm{d}\rho_0(w)\right),$$

where $u \in \mathbb{R}^d$ and $a_{w,t} \in \mathbb{R}^d$ for $w \in \mathbb{R}^d$ and $t \in \{1, \ldots, T\}$.

- Wasserstein optimal transportation map: $\mathcal{W} = \widetilde{\mathcal{W}} = \mathbb{R}^d$ and $f_W(x) = W(x)$. For random variables $X$ and $Y$ obeying distributions $P$ and $Q$ respectively: $\mathcal{W}^2(P, Q) = \min_{W:Q=f_W\sharp P} \mathrm{E}_{X \sim P}[\|X - f_W(X)\|^2]$.

## 3    Optimization error bound of transportation map learning

To show convergence of the dynamics (6), we utilize the recent result given by [45]. Let $\|W\|_\varepsilon := \left(\sum_{k \geq 0}(\mu_k)^{2\varepsilon}\langle W, e_k\rangle_\mathcal{H}^2\right)^{1/2}$ and $P_N W := \sum_{k=0}^{N-1}\langle W, e_k\rangle_\mathcal{H} e_k$ for $W \in \mathcal{H}$ where $(e_k)_k$ is the orthonormal system of $\mathcal{H}$. Accordingly, let $\mathcal{H}_N$ be the image of $P_N$: $\mathcal{H}_N = P_N\mathcal{H}$.

**Assumption 1.**

(i) (Eigenvalue condition) *There exists a constant $c_\mu$ such that $\mu_k \leq c_\mu(k + 1)^{-2}$.*

(ii) (Boundedness and Smoothness) *There exist $B, M > 0$ such that the gradient of the empirical risk is bounded by $B$ and is $M$-Lipschitz continuous with $\alpha \in (1/4, 1)$ almost surely:*

$$\|\nabla\widehat{\mathcal{L}}(W)\|_{\mathcal{H}} \le B \ (\forall W \in \mathcal{H}), \quad \|\nabla\widehat{\mathcal{L}}(W) - \nabla\widehat{\mathcal{L}}(W')\|_{\mathcal{H}} \le L\|W - W'\|_{\alpha} \ (\forall W, W' \in \mathcal{H}).$$

(iii) (Third order smoothness [13, Assumption 2.7]) *Let $\widehat{\mathcal{L}}_N : \mathcal{H}_N \to \mathbb{R}$ be $\widehat{\mathcal{L}}_N = \widehat{\mathcal{L}}(P_N W)$. $\widehat{\mathcal{L}}$ is three times differentiable, and there exists $\alpha' \in [0,1), C_{\alpha'} \in (0, \infty)$ such that for all $N \in \mathbb{N}$ and $\forall W, h, k \in \mathcal{H}_N, \|\nabla^3\widehat{\mathcal{L}}_N(W) \cdot (h, k)\|_{\alpha'} \le C_{\alpha'}\|h\|_{\mathcal{H}}\|k\|_{\mathcal{H}}, \ \|\nabla^3\widehat{\mathcal{L}}_N(W) \cdot (h, k)\|_{\mathcal{H}} \le C_{\alpha'}\|h\|_{-\alpha'}\|k\|_{\mathcal{H}}$ (a.s.), where $\nabla^3\widehat{\mathcal{L}}_N(W)$ is the third-order derivative, we identify it with third-order linear form, and we also write $\nabla^3\widehat{\mathcal{L}}_N(W) \cdot (h, k)$ for the Riesz representor of $l \in \mathcal{H} \mapsto \nabla^3\widehat{\mathcal{L}}_N(W) \cdot (h, k, l)$.*

The first condition controls the strength of the regularization term. The second condition ensures the smoothness of the loss function that yields the *disspativity* condition of the objective combined with the regularization term. That is, the solution of the gradient Langevin dynamics can remain a bounded region with high probability. The Lipschitz continuity of the gradient is a bit strong condition because the right hand side appears a weaker norm $\|\cdot\|_{\alpha}$ than the canonical norm $\|\cdot\|_{\mathcal{H}}$. However, this gives the geometric ergodicity (exponential convergence to the stationary distribution) of the discrete time dynamics. The third condition is more technical assumption. This condition is used for bounding the continuous time dynamics and discrete time dynamics. Intuitively, a smoother loss function makes the two dynamics closer. In particular, $\eta^{1/2-a}$ term appearing in the following bound can be shown by this condition.

Then, we can show the following weak convergence rate. Let $\pi_k$ be the probability measure on $\mathcal{H}$ corresponding to the distribution of $W_k$.

**Proposition 1.** *Assume Assumption 1 holds and $\beta > \eta$. Suppose that $\exists \bar{R} > 0, 0 \le \ell(Y, f_W(X)) \le \bar{R}$ for any $W \in \mathcal{H}$ (a.s.). Let $\rho = \frac{1}{1+\lambda\eta/\mu_0}$ and $b = \frac{\mu_0}{\lambda}B + \frac{c_\mu}{\beta\lambda}$. Then, for $\Lambda_\eta^* = \frac{\min\left(\frac{\lambda}{2\mu_0}, \frac{1}{2}\right)}{4\log(\kappa(\bar{V}+1)/(1-\delta))}\delta$ and $C_{W_0} = \kappa[\bar{V}+1] + \frac{\sqrt{2}(\bar{R}+b)}{\sqrt{\delta}}$ where $0 < \delta < 1$ satisfying $\delta = \Omega(\exp(-\Theta(\text{poly}(\lambda^{-1})\beta)))$, $\bar{b} = \max\{b, 1\}$, $\kappa = \bar{b}+1$ and $\bar{V} = 4\bar{b}/(\sqrt{(1+\rho^{1/\eta})/2}-\rho^{1/\eta})$ (where $\bar{V} = 4\bar{b}/(\sqrt{(1+\exp(-\frac{\lambda}{\mu_1}))/2}-\exp(-\frac{\lambda}{\mu_1}))$ for $\eta = 0$), and for any $0 < a < 1/4$, the following convergence bound holds for almost sure observation $D_n$: for either $L = \mathcal{L}$ or $L = \widehat{\mathcal{L}}$,*

$$|\mathbb{E}_{W_k \sim \pi_k}[L(W_k)] - \mathbb{E}_{W \sim \pi_\infty}[L(W)]| \le C_1\left[C_{W_0}\exp(-\Lambda_\eta^*\eta k) + \frac{\sqrt{\beta}}{\Lambda_0^*}\eta^{1/2-a}\right] =: \Xi_k, \quad (7)$$

*where $C_1$ is a constant depending only on $c_\mu, B, L, C_{\alpha'}, a, \bar{R}$ (independent of $\eta, k, \beta, \lambda$).*

We utilized the theories of [45] as the core technique to show this proposition. Its complete proof is given in Appendix A. We can see that as $k$ goes to infinity the first term of the right hand side converges exponentially, and as the step size $\eta$ goes to 0, the second term converges arbitrary close to the rate of $\sqrt{\eta}$. It is known that the convergence rate with respect to $\eta$ is optimal [15]. Therefore, if we choose sufficiently small $\eta$ and sufficiently large $k$, we can sample $W_k$ that obeys nearly the invariant measure $\pi_\infty$. As we will see later, sample from $\pi_\infty$ has a nice property in terms of generalization. As we have remarked in Remark 1, the convergence is guaranteed even for the finite width neural network setting, i.e., $\rho_0$ is a discrete distribution in the model (4). This is much advantageous against existing framework such as mean field analysis and NTK.

The above proposition gives a bound on the expectation of the loss of the solution $W_k$ instead of a high probability bound. However, due to the geometric ergodicity of the dynamics, by running the algorithm for sufficiently large steps, we can show that the probability that there *does not* appear $W_k$ in the trajectory that has a loss such that $L(W_k) - \mathbb{E}_{W \sim \pi_\infty}[L(W)] \le O(\Xi_k)$ approaches 0 with exponential rate. Since this direction requires much more involved mathematics, we consider a simpler one as described above.

## 4 Generalization error analysis

**Generalization gap bound** Here, we analyze the generalization error of the solution of $W_k$ obtained by the dynamics (6).

**Theorem 1.** *Assume Assumption 1 holds with $\beta > \eta$, and assume that the loss function is bounded, i.e., there exits $\bar{R} > 0$ such that $\forall W \in \mathcal{H}$, $0 \le \ell(Y, f_W(X)) \le \bar{R}$ (a.s.). Then, for any $1 > \delta > 0$, with probability $1 - \delta$, the generalization error is bounded by*

$$\mathrm{E}_{W_k}[\mathcal{L}(W_k)] \le \mathrm{E}_{W_k}[\widehat{\mathcal{L}}(W_k)] + \frac{\bar{R}^2}{\sqrt{n}}\left[2\left(1 + \frac{2\beta}{\sqrt{n}}\right) + \log\left(\frac{1 + e^{\bar{R}^2/2}}{\delta}\right)\right] + 2\Xi_k.$$

The proof is given in Appendix B. To prove this, we used a PAC-Bayes stability bound [52]. From this theorem, we have that the generalization error is bounded by $O(1/\sqrt{n})$ and the optimization error $\Xi_k$. The $O(1/\sqrt{n})$ term is the generalization gap for the stationary distribution, and as $k$ goes to infinity, the total generalization gap converges to this one. [44] also showed a PAC-Bayesian stability bound for a finite dimensional Langevin dynamics (roughly speaking, their bound is $O(\sqrt{\beta B^2/(n\lambda)})$), but their proof technique is quite different from ours. Our proof analyzes the generalization error under the stationary distribution of the dynamics and bounds the gap between the stationary distribution and the current solution, while [44] evaluated the bound by "accumulating" the error through the updates without analyzing the stationary distribution.

**Excess risk bound: fast learning rate**  Next, we bound the excess risk. Unlike the $O(1/\sqrt{n})$ convergence rate of the generalization gap bound, we can derive a fast learning rate which is faster than $O(1/\sqrt{n})$ in a setting of realizable case, i.e., a student-teacher model, for the excess risk instead of the generalization gap. As a concrete example, we keep the following two layer neural network model in our mind. For a map $W : \mathbb{R}^{d_1} \to \mathbb{R}^{d_2}$, let a "clipped map" $\bar{W}$ be $\bar{W}(w) := R \times \tanh(W(w)/R)$, where $R \ge 1$ is a constant and $\tanh$ is applied elementwise. Then, the following two layer neural network model falls into our analysis:

$$f_W(x) := \int_{\mathbb{R} \times \mathbb{R}^d} \bar{W}_2(a)\sigma(\bar{W}_1(w)^\top x)\mathrm{d}\rho_0(a, w) \tag{8}$$

for a measurable map $W = (W_1, W_2) : \mathbb{R}^d \times \mathbb{R} \to \mathbb{R}^d \times \mathbb{R}$ and an activation function $\sigma$ that is 1-Lipschitz continuous and included in a *Hölder class* $\mathcal{C}^3(\mathbb{R})$. Here, we used the clipping operation only for a technical reason because the current convergence analysis of the infinite dimensional Langevin dynamics requires a boundedness condition. This could be removed if we could show its convergence under more relaxed conditions. The fast learning rate analysis is not restricted to the two layer model, but it can be applied as long as the following statement is satisfied (e.g., ResNet).

**Lemma 1.** *For the model* (8), *if $\|x\| \le D$ for any $x \in \mathrm{supp}(P_X)$, then it holds that $\|f_W - f_{W'}\|_\infty \le (1 + RD)\|W - W'\|_{L_2(\rho_0)}$ where $\|W - W'\|^2_{L_2(\rho_0)} := \int \|W((a, w)) - W'((a, w))\|^2 \mathrm{d}\rho_0(a, w)$.*

The proof is given in Appendix C. This lemma indicates that to estimate a function $f_{W^*}$, its estimation error can be bounded by the estimation error of the parameter $W$. To ensure the smooth gradient assumption (Assumption 1-(ii)) and precisely characterize the estimation accuracy by the model complexity, we consider an RKHS with "smoothness" parameter $\gamma$ as the model of $W$. Let $T_K : \mathcal{H} \to \mathcal{H}$ be a linear bounded operator such that $\langle T_K h, h'\rangle_\mathcal{H} = \sum_{k=0}^\infty \mu_k \alpha_k \alpha'_k$ for $h = \sum_k \alpha_k e_k$ and $h' = \sum_k \alpha'_k e_k$. Let the range of power of $T_K$ be $\mathcal{H}_{K^\gamma} = \{f = T_K^{\gamma/2} h \mid h \in \mathcal{H}\}$ for $\gamma > 0$ which is equipped with the inner product $\langle h, h'\rangle_{\mathcal{H}_{K^\gamma}} = \sum_{k=0}^\infty \mu_k^{-\gamma}\alpha_k\alpha'_k$. We can see that $\gamma = 1$ corresponds to $\mathcal{H}_K$ and $\gamma$ controls the "complexity" of $\mathcal{H}_{K^\gamma}$, that is, if $\gamma < 1$, then $\mathcal{H}_K \hookrightarrow \mathcal{H}_{K^\gamma}$, and otherwise, $\mathcal{H}_{K^\gamma} \hookrightarrow \mathcal{H}_K$. We consider a problem of optimizing $\widehat{\mathcal{L}}(f_W)$ or $\mathcal{L}(f_W)$ with respect to $W$ in the model $\mathcal{H}_{K^\gamma}$. To so so, by noticing that any $g \in \mathcal{H}_{K^\gamma}$ can be written as $g = T_K^{\gamma/2}W$ for $W \in \mathcal{H}$, we write the empirical and population risk with respect to $W \in \mathcal{H}$ as $\widehat{\mathcal{L}}(W) = \widehat{\mathcal{L}}(f_{T_K^{\gamma/2}W})$, $\mathcal{L}(W) = \mathcal{L}(f_{T_K^{\gamma/2}W})$. Let $f^* \in \mathrm{argmin}_f \mathcal{L}(f)$ where min is taken over all measurable functions and we assume the existence of the minimizer.

**Assumption 2** (Bernstein condition and predictor condition [73, 7])**.** *The Bernstein condition is satisfied: there exist $C_B > 0$ and $s \in (0, 1]$ such that for any $f_W$ ($W \in \mathcal{H}$),*

$$\mathrm{E}[(\ell(Y, f_W(X)) - \ell(Y, f^*(X)))^2] \le C_B(\mathcal{L}(f_W) - \mathcal{L}(f^*))^s.$$

*Moreover, we assume that, for any $h : \mathbb{R}^d \to \mathbb{R}$ and $x \in \mathrm{supp}(P_X)$, it holds that*

$$\mathrm{E}_{Y|X=x}\left[\exp\left(-\frac{\beta}{n}(\ell(Y, h(x)) - \ell(Y, f^*(x)))\right)\right] \le 1.$$

The first assumption is called *Bernstein condition*. We can show that this condition is satisfied by the logistic loss and the squared loss with bounded $f_W$ and $f^*$ (Theorem 3). The second assumption is called *predictor condition* [73] and can be satisfied if $\ell$ is a log-likelihood function and the model is correctly specified (that is, the true conditional probability density (or probability mass) $p(y|x)$ is expressed as $p(y|x) \simeq \exp(-\ell(y, f^*(x)))$). To extend the theory to misspecified situations, we need the second assumption. For example, if we use a squared loss in a regression problem whereas the label noise is *not* Gaussian, then it is a misspecified situation but if the noise has a light tail (such as sub-Gaussian), then the assumption can be satisfied [73].

Our analysis is valid even if $f^*$ cannot be represented by $f_W$ for $W \in \mathcal{H}$. This model misspecification can be incorporated as bias-variance trade-off in the excess risk bound. This trade-off can be captured by the following *concentration function*. Let $\mathcal{H}_{\tilde{K}} = \mathcal{H}_{K^{\gamma+1}}$, and the Gaussian process law of $T_K^{\gamma/2}W$ for $W \sim \nu_\beta$ be $\tilde{\nu}_\beta$. Then, define the concentration function as

$$\phi_{\beta,\lambda}(\epsilon) := \inf_{h \in \mathcal{H}_{\tilde{K}}: \mathcal{L}(h) - \mathcal{L}(f^*) \leq \epsilon^2} \beta\lambda\|h\|_{\mathcal{H}_{\tilde{K}}}^2 - \log \tilde{\nu}_\beta(\{h \in \mathcal{H} : \|h\|_{\mathcal{H}} \leq \epsilon\}) + \log(2),$$

where, if there does not exist $h \in \mathcal{H}_{\tilde{K}}$ satisfying the condition in inf, then we set $\phi_{\beta,\lambda}(\epsilon) = \infty$.

**Theorem 2.** *Assume that Assumption 2 holds, $\|x\| \leq D$ $(\forall x \in \mathcal{X})$, $\gamma > 1/2$, $\beta > \eta$ and $\beta \leq n$. Assume that the loss function $\ell(y, \cdot)$ is included in $\mathcal{C}^3(\mathbb{R})$ for any $y \in \text{supp}(P_Y)$ and there exists $B > 0$ such that $|\frac{\partial^k}{\partial u^k}\ell(y, u)| \leq B$ $(\forall u \in \mathbb{R}$ s.t. $|u| \leq R, \forall y \in \text{supp}(P_Y), k = 1, 2, 3)$. Assume also that $0 \leq \ell(Y, f(X)) \leq \bar{R}$ (a.s.) for any $f = f_W$ $(W \in \mathcal{H})$ and $f = f^*$, and $\bar{\ell}_x(u) := \mathrm{E}_{Y|X=x}[\ell(Y, u)]$ satisfies $|\frac{d\bar{\ell}_x}{du}(u) - \frac{d\bar{\ell}_x}{du}(u')| \leq L|u - u'|$ $(\forall u, u' \in \mathbb{R}, \forall x \in \mathcal{X})$ for a constant $L > 0$. Let $\tilde{\alpha} := 1/\{2(\gamma+1)\}$ and $\theta$ be an arbitrary real number satisfying $0 < \theta < 1 - \tilde{\alpha}$. We define $\epsilon^* := \inf\{\epsilon > 0 : \phi_{\beta,\lambda}(\epsilon) \leq \beta\epsilon^2\} \vee n^{-\frac{1}{2-s}}$. Then, the expected excess risk is bounded as*

$$\mathrm{E}_{D^n}\left[\mathrm{E}_{W_k}[\mathcal{L}(W_k)] - \mathcal{L}(f^*)\right] \leq C\left[\epsilon^{*2} \vee \left(\frac{\beta}{n}\epsilon^{*2} + n^{-\frac{1}{1+\tilde{\alpha}/\theta}}(\lambda\beta)^{\frac{2\tilde{\alpha}/\theta}{1+\tilde{\alpha}/\theta}}\right)^{\frac{1}{2-s}} \vee \frac{1}{n}\right] + \Xi_k, \quad (9)$$

*where $C$ is a constant independent of $n, \beta, \lambda, \eta, k$.*

The proof is given in Appendix D.2. It is proven by using the technique of nonparametric Bayes contraction rate analysis [25, 71, 72]. However, we cannot adapt these existing techniques because (i) the loss function is not necessarily the log-likelihood function, (ii) the inverse temperature is generally different from the sample size. In that sense, our proof is novel to derive an excess risk for (i) a misspecified model, and (ii) a randomized estimator with a general inverse temperature parameter.

The bound is about expectation of the excess risk instead of high probability bound. However, a high probability bound is also provided in the proof and the expectation bound is derived from the high probability bound.

If the bias is not zero, i.e., $\inf_{W \in \mathcal{H}} \mathcal{L}(W) - \mathcal{L}(f^*) = \delta_0 > 0$, then we may choose $\epsilon^{*2} = \Theta(\delta_0)$ because $\phi_{\beta,\lambda}(\epsilon)$ is finite for $\epsilon^2 > \delta_0$ and infinite for $\epsilon^2 < \delta_0$. Thus, a misspecified setting is covered.

**(i) Example of fast rate: Regression** Here, we apply our general result to a nonparametric regression problem by the neural network model. We consider the following nonparametric regression model: $y_i = f_{W^*}(x_i) + \epsilon_i$, for $W^* \in \mathcal{H}$ where $\epsilon_i$ is an i.i.d. noise with mean 0 and $|\epsilon_i| \leq C < \infty$ (a.s.). To estimate $f_{W^*}$, we employ the squared loss $\ell(y, f) = (y - f)^2$. Then, we can easily confirm that $f^*$ is achieved by $f_{W^*}$ via a simple calculation: $\text{argmin}_f \mathcal{L}(f) = f_{W^*}$. Moreover, for the squared loss, $s = 1$ is satisfied as remarked just after Assumption 2. Moreover, we further assume that $W^* \in \mathcal{H}_{K^{\theta(\gamma+1)}}$ for $\theta < 1 - \tilde{\alpha}$. Then, the "bias" and "variance" terms can be evaluated as $\inf_{h \in \mathcal{H}_{\tilde{K}}: \mathcal{L}(h) - \mathcal{L}(f^*) \leq \epsilon^2} \lambda_\beta\|h\|_{\mathcal{H}_{\tilde{K}}}^2 \lesssim \lambda\beta\epsilon^{-\frac{2(1-\theta)}{\theta}}$ and $-\log \tilde{\nu}_\beta(\{h \in \mathcal{H} : \|h\|_{\mathcal{H}} \leq \epsilon\}) \lesssim (\epsilon/(\lambda\beta)^{1/2})^{-\frac{2\tilde{\alpha}}{1-\tilde{\alpha}}}$. Accordingly, we can show the following excess risk bound:

$$\mathrm{E}_{D^n}\left[\mathrm{E}_{W_k}[\mathcal{L}(W_k)] - \mathcal{L}(f^*)\right] \lesssim \max\left\{(\lambda\beta)^{\frac{2\tilde{\alpha}/\theta}{1+\tilde{\alpha}/\theta}}n^{-\frac{1}{1+\tilde{\alpha}/\theta}}, \lambda^{-\tilde{\alpha}}\beta^{-1}, \lambda^\theta, 1/n\right\} + \Xi_k, \quad (10)$$

(see Appendix D.4 for the derivation). In particular, if $\beta = \lambda^{-1} = n$, then this convergence rate can be rewritten as $\max\{n^{-\frac{1}{1+\tilde{\alpha}/\theta}}, n^{-\theta}\} = n^{-\theta}$ ($\because \theta < 1 - \tilde{\alpha}$), which can be faster than $1/\sqrt{n}$ and is controlled by the "difficulty" of the problem $\tilde{\alpha}$ and $\theta$.

**Remark 2.** *As an example, if the RKHS $\mathcal{H}_K$ is a Sobolev space $W_2^{a+d/2}(\mathbb{R}^d)$ with regularity parameter $a + d/2$ (more precisely, each output $W_i(\cdot)$ is a member of a Sobolev space) and $\mathcal{H}$ is $L_2(\rho_0)$, then we can set $\tilde{\alpha} = \frac{d}{2a+d}$. If the true parameter $W^*$ is included in another Sobolev space $W_2^b(\mathbb{R}^d)$ for $b \leq a$, then we may choose $\theta = 2b/(2a+d)$ and the convergence rate is bounded by $n^{-2b/(2a+d)}$, which coincides with the posterior contraction rate of Gaussian process estimator derived in [72]. It is known that, if $a = b$, this achieves the* minimax optimal rate *[78].*

**(ii) Example of fast rate: Classification (exponential convergence)**   Here, we consider a binary classification problem $y \in \{\pm 1\}$. We employ the logistic loss function $\ell(y, f) = \log(1 + \exp(-yf))$ for $y \in \{\pm 1\}$ and $f \in \mathbb{R}$. Corresponding to the loss function, we define the expected loss conditioned by $X = x$ as $h(u|x) = \mathrm{E}[\ell(Y, u)|X = x]$. Note that $h(0|x) = \log(2)$. We assume that the strength of noise of this binary classification problem is low as follows.

**Assumption 3** (Strong low noise condition). *Let $h^*(x) := \inf_{u \in \mathbb{R}} h(u|x)$. Assume that there exists $\delta > 0$ such that $h^*(x) \leq \log(2) - \delta$ ($\forall x \in \mathcal{X}$). Moreover, there exists $W^* \in \mathcal{H}$ such that $f^* = f_{W^*}$, that is, $\sup_{x \in \mathrm{supp}(P_X)} |h(f_{W^*}(x)|x) - h^*(x)| = 0$.*

The first assumption is satisfied if the label probability is away from the even probability $1/2$: $|P(Y|X = x) - 1/2| > \Omega(\sqrt{\delta})$. This condition means that the class label has less noisy than completely random labeling. In that sense, we call this assumption the *strong low noise condition*, which has been analyzed in [36, 4, 49]. A weaker low noise condition was introduced by [70] as Tsybakov's low noise condition. The second assumption can be relaxed to the existence of $W$ only for some $\epsilon > c_0\delta$ with sufficiently small $c_0$, but we don't pursuit this direction for simplicity.

**Assumption 4.** *Assume $\mathcal{X}(= \mathrm{supp}(P_X)) \subset [0, 1]^d$ and $\mathcal{X}$ is a minimally smooth domain in a sense of [61]. $P_X$ has a density $p(x)$ which is lower bounded as $p(x) \geq c_0$ ($\forall x \in \mathrm{supp}(P_X)$) on its support. For $2m > d$ and $m \geq 3$, the activation function satisfies $\sigma \in \mathcal{C}^m(\mathbb{R})$ and $f^*$ is included in the Sobolev space $W_2^m(\mathcal{X})$ defined on $\mathcal{X}$ (see [21] for its definition).*

The following theorem gives an upper-bound of the probability of "perfect classification" for the estimator. More specifically, it shows the error probability converges in an *exponential rate*.

**Theorem 3.** *Under Assumptions 3 and 4, the convergence in Theorem 2 holds for $s = 1$. Let $g^*(x) = \mathrm{sign}(P(Y = 1|X = x) - 1/2)$ be the Bayes classifier. If the sample size $n$ is sufficiently large and $\lambda, \beta$ are appropriately chosen, then the classification error converges exponentially with respect to $\beta$ and $k$:*

$$\mathrm{E}[\pi_k(\{W_k \in \mathcal{H} \mid P_X(\mathrm{sign}(f_{W_k}(X)) = g^*(X)) \neq 1\})] \lesssim \frac{\Xi_k}{\delta^{2m/(2m-d)}} + \exp(-c'\beta\delta^{\frac{2m}{2m-d}}).$$

The proof is given in Appendix D.3. This theorem states that if we choose the step size $\eta$ sufficiently small, then the error probability converges exponentially as $k$ and $\beta$ increase. Even if the first term of the right hand side is larger than the second term, we can make this as small as the second term by running the algorithm several times and picking up the best one with respect to validation error if $\Xi_k \ll 1$ (see Appendix D.3 for this discussion).

## 5   Conclusion

In this paper, we have formulated the deep learning training as a transportation map estimation and analyzed its convergence and generalization error through the infinite dimensional Langevin dynamics. Unlike exiting analysis, our formulation can incorporate spatial correlation of noise and achieve global convergence without taking the limit of infinite width. The generalization analysis reveals the dynamics achieves a stable estimator with $O(1/\sqrt{n})$ convergence of generalization error and shows fast learning rate of the excess risk. Finally, we have shown a convergence rate of excess risk for regression and classification. The rate for regression recovers the minimax optimal rate known in Bayesian nonparametrics and that for classification achieves exponential convergence under the strong low noise condition.

## Broader impact

**Benefit** Since deep learning is used in several applications across broad range of areas, our theoretical analysis about optimization of deep learning would influence wide range of areas in terms of understanding of the algorithmic behavior. One of the biggest criticisms on deep learning is its poor explainability and interpretability. Our work on optimization analysis of deep learning can much improve explainability and would facilitate its usage. This is quite important step toward trustworthy machine learning.

**Potential risk** On the other hand, this is purely theoretical work and thus would not directly bring on severe ethical issues. However, misunderstanding of theoretical work would cause misuse of its statement to conduct an intensional opinion making. To avoid such a potential risk, we made our best effort to minimize technical ambiguity in our paper presentation.

## Acknowledgment

I would like to thank Atsushi Nitanda for insightful comments. TS was partially supported by JSPS KAKENHI (18K19793,18H03201, and 20H00576), Japan Digital Design, and JST CREST.

## Footnotes

[1]More natural modeling would be that the regularization $A$ and the covariance of $\xi_t$ depend on the current solution $W_t$, but we consider this simplest model for technical tractability.

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
