[Supplementary Material]

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

[2]For a metric space $\tilde{\mathcal{F}}$ equipped with a metric $\tilde{d}$, the $\epsilon$-covering number $\mathcal{N}(\tilde{\mathcal{F}}, \tilde{d}, \epsilon)$ is defined as the minimum number of balls with radius $\epsilon$ (measured by the metric $\tilde{d}$) to cover the metric space $\tilde{\mathcal{F}}$.

[3]This is a more precise meaning of the sentence "the sample size $n$ is sufficiently large and $\lambda$ is appropriately chosen" in the statement.

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

# ——Appendix——

## A  Proof of Proposition 1

We apply the result [45]. Let $\{Z_n\}_{n \in \mathbb{N}}$ be a dynamics obeying

$$Z_{n+1} = S_\eta Z_n + \sqrt{\eta/\beta} S_\eta \varepsilon_n,$$

with $Z_0 = 0$. Let $k(p) := \sup_{n \geq 0} \mathrm{E}(\|Z_n\|^p)$ for $p > 0$, then it is known that $k(p) < \infty$ for any $p > 0$. Let $\{Z_n\}_{n \in \mathbb{N}}$ solve $\overline{Z}_0 = 0$ and with $\beta > \eta$. Then, we can show that $k(p) := \sup_{n \geq 0} \mathrm{E}(\|Z_n\|^p) < \infty \ (\forall p > 0)$[45]. Using $k(p)$, we define $b' = \frac{\mu_0}{\lambda} B + k(1)$. We will show that $k(1) \leq \frac{c_\mu}{\beta \lambda}$. Then, we can see that $b' \leq b$. Now, we show $k(1) \leq \frac{c_\mu}{\beta \lambda}$. First, note that

$$Z_n = \sqrt{\frac{\eta}{\beta}} \sum_{\ell=0}^{n-1} S_\eta^{n-\ell} \epsilon_\ell.$$

Therefore, we have

$$\mathrm{E}[\|Z_n\|^2] = \frac{\eta}{\beta} \sum_{\ell=0}^{n-1} \mathrm{Tr}[S_\eta^{2(n-\ell)}] = \frac{\eta}{\beta} \mathrm{Tr}\left[(S_\eta^2 - S_\eta^{2n})(I - S_\eta^2)^{-1}\right] \leq \frac{\eta}{\beta} \mathrm{Tr}\left[S_\eta^2(I - S_\eta^2)^{-1}\right]$$

$$= \frac{\eta}{\beta} \sum_{k=0}^{\infty} \left(\frac{1}{1 + \eta\lambda/\mu_k}\right)^2 \left(1 - \frac{1}{(1 + \eta\lambda/\mu_k)^2}\right)^{-1} = \frac{\eta}{\beta} \sum_{k=0}^{\infty} \frac{1}{(1 + \eta\lambda/\mu_k)^2 - 1}$$

$$\leq \frac{\eta}{\beta} \sum_{k=0}^{\infty} \frac{1}{2\eta\lambda/\mu_k} \leq \frac{1}{2\beta\lambda} \sum_{k=0}^{\infty} c_\mu (k+1)^{-2} \leq \frac{c_\mu}{\beta \lambda}.$$

Then, Jensen's inequality yields $k(1) = \mathrm{E}[\|Z_n\|_{\mathcal{H}}] \leq \sqrt{\mathrm{E}[\|Z_n\|_{\mathcal{H}}^2]} \leq \sqrt{\frac{c_\mu}{\beta\lambda}}$.

Let $\phi : \mathcal{H} \to \mathbb{R}$ be a test function satisfying $|\phi(\cdot)| \leq V(\cdot)$ and $\|\phi(x) - \phi(y)\| \leq M\|x - y\| \ (x, y \in \mathcal{H})$ for $M > 0$. Then, [45] showed that there exists a unique invariant measure $\mu_\eta$ and the following exponential convergence of the expectation of $\phi$ holds:

$$|\mathrm{E}_{x_0}[\phi(X_n)] - \mathrm{E}[\phi(X^{\mu_\eta})]| \leq C_{x_0} \exp(-\Lambda_\eta^*(\eta n - 1)). \tag{11}$$

where

$$\Lambda_\eta^* = \frac{\min\left(\frac{\lambda}{2\mu_0}, \frac{1}{2}\right)}{4 \log(\kappa(\bar{V} + 1)/(1 - \delta))} \delta, \ C_{W_0} = \kappa[\bar{V} + 1] + \frac{\sqrt{2}(\bar{R} + b)}{\sqrt{\delta}}$$

with $0 < \delta < 1$ satisfying $\delta = \Omega(\exp(-C'\mathrm{poly}(\lambda^{-1})\beta))$, $\bar{b} = \max\{b, 1\}$, $\kappa = \bar{b} + 1$ and $\bar{V} = \frac{4\bar{b}}{\sqrt{(1 + \rho^{1/\eta})/2 - \rho^{1/\eta}}}$. To show this we note that $\beta$ in [45] is $2\beta$ using $\beta$ in this paper. The definition of $\delta$ is not explicitly shown in [45] (in particular, $\lambda$ is omitted), but we can recover our definition from the proof. Moreover, [45] assumed that there exists $\lambda_0, C_{\alpha,2} \in (0, \infty)$ such that

$$\|\nabla\widehat{\mathcal{L}}(W) - \nabla\widehat{\mathcal{L}}(W')\|_{\mathcal{H}} \leq L\|W - W'\|_{\mathcal{H}} \ (\forall W, W' \in \mathcal{H}),$$

$$|\nabla^2\widehat{\mathcal{L}}(W) \cdot (h, k)| \leq C_{\alpha,2}\|h\|_{\mathcal{H}}\|k\|_\alpha \ (\forall W, h, k \in \mathcal{H}),$$

instead of our assumption $\|\nabla\widehat{\mathcal{L}}(W) - \nabla\widehat{\mathcal{L}}(W')\|_{\mathcal{H}} \leq L\|W - W'\|_\alpha$. However, we can see that their proof is valid even under our assumption.

Let $C_b^2$ be a set of functions $f : \mathcal{H} \to \mathbb{R}$ that is continuously twice differentiable with bounded derivatives. Under the same setting above, [45] also showed that, for any $0 < \kappa < 1/2$, $0 < \eta_0$, there exists a constant $C$ such that, if the test function $\phi$ satisfies $\phi \in C_b^2$, then for any $0 < \eta < \eta_0$, it holds that

$$|\mathrm{E}[\phi(X^{\mu_\eta})] - \mathrm{E}[\phi(X^{\pi_\infty})]| \leq C \frac{\|\phi\|_{0,2}}{\Lambda_0^*} c_\beta \eta^{1/2 - \kappa}, \tag{12}$$

where $\|\phi\|_{0,2} := \max\{\|\phi\|_\infty, \sup_{x \in \mathcal{H}} \|\nabla \phi(x)\|_{\mathcal{H}}, \sup_{x \in \mathcal{H}} \|\nabla^2 \phi(x)\|_{\mathcal{B}(\mathcal{H})}\}$ for $\phi \in C_b^2$ where $\|\cdot\|_{\mathcal{B}(\mathcal{H})}$ is the norm as a linear operator.

Thus, if we let $\phi(\cdot) = \widehat{\mathcal{L}}(\cdot)/\bar{R}$, then $\phi$ satisfies the assumption with $M = B/\bar{R}$. Therefore, we obtain that

$$|\mathrm{E}_{x_0}[\widehat{\mathcal{L}}(X_n)] - \mathrm{E}[\widehat{\mathcal{L}}(X^{\pi_\infty})]| \le \bar{R}\left[C_{x_0}\exp(-\Lambda_\eta^*(\eta n - 1)) + C\frac{\|\phi\|_{0,2}}{\Lambda_0^*}c_\beta \eta^{1/2-\kappa}\right].$$

This gives the assertion.

Finally, we would like to note that since the assumption is satisfied almost surely, $\mathcal{L}$ also satisfies the assumption instead of $\widehat{\mathcal{L}}$. That means the same convergence rate holds also for $\phi(W) = \mathcal{L}$.

## B  Proof of Theorem 1

*Proof.* [52] proved that, for any probability measure $Q$ which is absolutely continuous to $\pi_\infty$, it holds that

$$\mathrm{E}_{W \sim Q}[\mathcal{L}(W)] \le \mathrm{E}_{W \sim Q}[\widehat{\mathcal{L}}(W)] + \frac{1}{\sqrt{n}}\mathrm{KL}(Q\|\pi_\infty) + \frac{\bar{R}^2}{\sqrt{n}}\left[2\left(1 + \frac{2\beta}{\sqrt{n}}\right) + \log\left(\frac{1 + e^{\bar{R}^2/2}}{\delta}\right)\right],$$
(13)

with probability $1 - \delta$.

On the other hand, Proposition 1 gives that

$$|\mathrm{E}_{W_k \sim \pi_k}[\mathcal{L}(W_k)] - \mathrm{E}_{W \sim \pi_\infty}[\mathcal{L}(W)]| \le \Xi_k,$$
$$|\mathrm{E}_{W_k \sim \pi_k}[\widehat{\mathcal{L}}(W_k)] - \mathrm{E}_{W \sim \pi_\infty}[\widehat{\mathcal{L}}(W)]| \le \Xi_k.$$

Then, by substituting $Q = \pi_\infty$ into Eq. (13) and applying the two inequalities above, we obtain the assertion.  $\square$

## C  Proof of Lemma 1

By the definition of $f_W$, we have

$f_W(x) - f_{W'}(x)$
$$\le \int \left[(\bar{W}_2(a) - \bar{W}_2'(a))\sigma(\bar{W}_1(w)^\top x) + \bar{W}_2'(a)(\sigma(\bar{W}_1(w)^\top x) - \sigma(\bar{W}_1'(w)^\top x))\right] \mathrm{d}\rho_0(a, w)$$
$$\le \sqrt{\int (\bar{W}_2(a) - \bar{W}_2'(a))^2 \mathrm{d}\rho_0(a, w)} + \sqrt{\int (\bar{W}_2'(a))^2(\sigma(\bar{W}_1(w)^\top x) - \sigma(\bar{W}_1'(w)^\top x))^2 \mathrm{d}\rho_0(a, w)}$$
$$\le \sqrt{\int (W_2(a) - W_2'(a))^2 \mathrm{d}\rho_0(a, w)} + R\sqrt{\int (\sigma(\bar{W}_1(w)^\top x) - \sigma(\bar{W}_1'(w)^\top x))^2 \mathrm{d}\rho_0(a, w)},$$

where we used $|\sigma(x)| \le 0$, 1-Lipschitz continuity of the clipping operation and $|\bar{W}_2'(a)| \le R$. By noticing that $\|x\| \le D$ and $\sigma$ and the clipping operation $W(w) \mapsto \bar{W}(w)$ are 1-Lipschitz continuous, the right hand side can be further bounded by

$$\sqrt{\int (W_2(a) - W_2'(a))^2 \mathrm{d}\rho_0(a, w)} + R\sqrt{\int (\bar{W}_1(w)^\top x - \bar{W}_1'(w)^\top x)^2 \mathrm{d}\rho_0(a, w)}$$
$$\le \sqrt{\int (W_2(a) - W_2'(a))^2 \mathrm{d}\rho_0(a, w)} + RD\sqrt{\int \|\bar{W}_1(w) - \bar{W}_1'(w)\|^2 \mathrm{d}\rho_0(a, w)}$$
$$\le \sqrt{\int (W_2(a) - W_2'(a))^2 \mathrm{d}\rho_0(a, w)} + RD\sqrt{\int \|W_1(w) - W_1'(w)\|^2 \mathrm{d}\rho_0(a, w)}$$

$$\leq (1 + RD)\sqrt{\int (W_2(a) - W_2'(a))^2 + \|W_1(w) - W_1'(w)\|^2 \mathrm{d}\rho_0(a, w)}$$

$$\leq (1 + RD)\sqrt{\int \|W((a, w)) - W'((a, w))\|^2 \mathrm{d}\rho_0(a, w)} = (1 + RD)\|W - W'\|_{L_2(\rho_0)}.$$

This gives the assertion.

# D Proof of fast rate of excess risk bounds

## D.1 Gaussian correlation inequality

**Lemma 2** (Gaussian correlation inequality). *Let $\tilde{\mathcal{H}}$ be a separable Hilbert space equipped with the complete orthonormal system $(e_i)_{i=1}^\infty$, and suppose that $\nu$ is a Gaussian measure in $\tilde{\mathcal{H}}$ with mean 0 and covariance $\Sigma = \mathrm{diag}\,(\mu_1, \mu_2, \dots)$ with respect to CONS $(e_i)_i$ where $\sum_{i=1}^n \mu_i^2 < \infty$, that is, $\nu$ is the distribution corresponding to $\sum_{i=1}^\infty \xi_i \sqrt{\mu_i} e_i$ for $\xi_i \sim N(0, 1)$ (i.id.). Let $\mathcal{A}^1 = \{\sum_{i=1}^\infty \alpha_i e_i \in \tilde{\mathcal{H}} \mid \sum_{i=1}^\infty a_i \alpha_i^2 \leq 1, \alpha_i \in \mathbb{R}\}$ for $a_i \geq 0$ $(i = 1, 2, \dots)$ and $\mathcal{A}^2 = \{\sum_{i=1}^\infty \alpha_i e_i \in \tilde{\mathcal{H}} \mid \sum_{i=1}^\infty b_i \alpha_i^2 \leq 1, \alpha_i \in \mathbb{R}\}$ for $b_i \geq 0$ $(i = 1, 2, \dots)$. Then, we have*

$$\nu(\mathcal{A}^1 \cap \mathcal{A}^2) \geq \nu(\mathcal{A}^1)\nu(\mathcal{A}^2).$$

*Proof.* Let $\mathcal{A}_n^1$ an $\mathcal{A}_n^2$ be the cylinder set that "truncates" $\mathcal{A}^1$ an $\mathcal{A}^2$ up to index $n$: $\mathcal{A}_n^1 = \{\sum_{i=1}^\infty \alpha_i e_i \in \mathcal{H} \mid \sum_{i=1}^n a_i \alpha_i^2 \leq 1\}$ and $\mathcal{A}_n^2 = \{\sum_{i=1}^\infty \alpha_i e_i \in \mathcal{H} \mid |\sum_{i=1}^n b_i \alpha_i^2 \leq 1\}$. By the Gaussian correlation inequality [55, 38], it holds that

$$\nu(\mathcal{A}_n^1 \cap \mathcal{A}_n^2) \geq \nu(\mathcal{A}_n^1)\nu(\mathcal{A}_n^2).$$

Note that we can apply the Gaussian correlation inequality for a finite dimensional Gaussian measure. Next, we extend this inequality to the infinite dimensional space. Since $(\mathcal{A}_n^1)_n$ is a monotonically decreasing sequence, i.e., $\mathcal{A}_n^1 \subseteq \mathcal{A}_m^1$ for $m < n$, and $\cap_{n=1}^\infty \mathcal{A}_n^1 = \mathcal{A}^1$, the continuity of probability measure gives that $\lim_{n \to \infty} \nu(\mathcal{A}_n^1) = \nu(\mathcal{A}^1)$. Similarly, it holds that $\lim_{n \to \infty} \nu(\mathcal{A}_n^2) = \nu(\mathcal{A}^2)$.

Since $\mathcal{A}^2 \subset \mathcal{A}_n^1$ and $\mathcal{A}^2 \subset \mathcal{A}_n^2$, it holds that $\nu(\mathcal{A}^1 \cap \mathcal{A}^2) \leq \nu(\mathcal{A}_n^1 \cap \mathcal{A}_n^2)$. On the other hand, we also have

$$\nu(\mathcal{A}_n^1 \cap \mathcal{A}_n^2) = \nu((\mathcal{A}^1 \cup (\mathcal{A}_n^1 \backslash \mathcal{A}^1)) \cap (\mathcal{A}^2 \cup (\mathcal{A}_n^2 \backslash \mathcal{A}^2)))$$
$$\leq \nu((\mathcal{A}^1 \cap \mathcal{A}^2) \cup (\mathcal{A}_n^1 \backslash \mathcal{A}^1) \cup (\mathcal{A}_n^2 \backslash \mathcal{A}^2))$$
$$\leq \nu(\mathcal{A}^1 \cap \mathcal{A}^2) + \nu(\mathcal{A}_n^1 \backslash \mathcal{A}^1) + \nu(\mathcal{A}_n^2 \backslash \mathcal{A}^2)$$
$$\to \nu(\mathcal{A}^1 \cap \mathcal{A}^2).$$

Therefore, we have that

$$\lim_{n \to \infty} \nu(\mathcal{A}_n^1 \cap \mathcal{A}_n^2) = \nu(\mathcal{A}^1 \cap \mathcal{A}^2).$$

Combining all these arguments, we finally have that

$$\nu(\mathcal{A}^1 \cap \mathcal{A}^2) \geq \nu(\mathcal{A}^1)\nu(\mathcal{A}^2).$$

$\square$

## D.2 Proof of general excess risk bound (Theorem 2)

*Proof.* Since $\gamma > 1/2$, $\sigma$ and $\ell(y, \cdot)$ are in $\mathcal{C}^3(\mathbb{R})$ where $\ell$ has a bounded partial derivative on a bounded domain, we can easily verify that the empirical risk satisfies Assumption 1 by noticing the clipping operation in the model.

For $0 < \theta < 1$, let $\mathcal{H}_{\tilde{K}^\theta} := \mathcal{H}_{K^{\theta(\gamma+1)}}$. It is known that if the natural inclusion $I_{\tilde{K}, \tilde{K}^\theta} : \mathcal{H}_{\tilde{K}} \to \mathcal{H}_{\tilde{K}^\theta}$ is Hilbert-Schmidt, then the sample path of $\tilde{\nu}_\beta$ is included in $\mathcal{H}_{\tilde{K}^\theta}$ probability 1 (Theorem 5.2 of [62]). In our case, since $\mu_k \lesssim 1/k^2$, the eigenvalues $(\mu_k(\tilde{K}))_{k=1}^\infty$ of $\tilde{K}$ satisfies

$\mu_k(\tilde{K}) \lesssim 1/k^{2(\gamma+1)}$. Theorem 5.2 of [62] also states that $I_{\tilde{K}, \tilde{K}^\theta}$ is Hilbert-Schmidt if and only if $\sum_{k=0}^{\infty} \mu_k(\tilde{K})^{1-\theta} < \infty$. Therefore, by setting $\tilde{\alpha} := 1/\{2(\gamma+1)\}$, $\theta < 1 - \tilde{\alpha}$ is sufficient for this property. From now on, we assume that $\theta < 1 - \tilde{\alpha}$. For notational simplicity, let $\lambda_\beta := \lambda\beta$.

By definition, we have

$$\mathrm{E}_{W \sim \tilde{\nu}_\beta}[\|T_K^{\gamma/2} W\|_{\mathcal{H}_{\tilde{K}^\theta}}^2] = \mathrm{Tr}[T_K^{\gamma/2}(\lambda_\beta^{-1} T_K) T_K^{\gamma/2} T_K^{-\theta(\gamma+1)}] = \lambda_\beta^{-1} \mathrm{Tr}[T_K^{(\gamma+1)(1-\theta)}]$$

Note that the assumption $\theta < 1 - \tilde{\alpha}$ ensures the right hand side is finite. Therefore, we obtain that, for $\bar{R}_\theta > 0$,

$$\tilde{\nu}_\beta(\{h \in \mathcal{H} \mid \|h\|_{\mathcal{H}_{\tilde{K}^\theta}} \geq \lambda_\beta^{-1/2} \bar{R}_\theta\}) \leq \frac{\mathrm{E}_{W \sim \tilde{\nu}_\beta}[\|T_K^{\gamma/2} W\|_{\mathcal{H}_{\tilde{K}^\theta}}^2]}{\lambda_\beta^{-1} \bar{R}_\theta^2} \leq \frac{\lambda_\beta^{-1} \mathrm{Tr}[T_K^{(\gamma+1)(1-\theta)}]}{\lambda_\beta^{-1} \bar{R}_\theta^2}$$

$$= \frac{\mathrm{Tr}[T_K^{(\gamma+1)(1-\theta)}]}{\bar{R}_\theta^2}.$$

Hence, by setting $\bar{R}_\theta = \sqrt{2\mathrm{Tr}[T_K^{(\gamma+1)(1-\theta)}]}$, we can guarantee that $\tilde{\nu}_\beta(\{h \in \mathcal{H} \mid \|h\|_{\mathcal{H}_{\tilde{K}^\theta}} \leq \lambda_\beta^{-1/2} \bar{R}_\theta\}) \geq 1/2$.

Let $B_\epsilon = (\epsilon \mathcal{B}_\mathcal{H}) \cap (\lambda_\beta^{-1/2} \bar{R}_\theta \mathcal{B}_{\mathcal{H}_{\tilde{K}^\theta}})$. We define

$$\phi_{\beta,\lambda}^{(0)}(\epsilon) := -\log \tilde{\nu}_\beta(\{W \in \mathcal{H} : W \in B_\epsilon\}).$$

For any $\delta > 0$, pick up $h^* \in \mathcal{H}_{\tilde{K}}$ that satisfies

$$\lambda_\beta \|h^*\|_{\mathcal{H}_{\tilde{K}}}^2 \leq (1+\delta) \inf_{h \in \mathcal{H}_{\tilde{K}} : \mathcal{L}(h) - \mathcal{L}(f^*) \leq \epsilon^2} \lambda_\beta \|h\|_{\mathcal{H}_{\tilde{K}}}^2$$

$$\mathcal{L}(f_{h^*}) - \mathcal{L}(f^*) \leq \epsilon^2.$$

Then, by Borel's inequality, it holds that

$$-\log \tilde{\nu}_\beta(\{h \in \mathcal{H} : \|h - h^*\|_\mathcal{H} \leq \epsilon\}) \leq (1+\delta)\phi_{\beta,\lambda}(\epsilon).$$

By the smoothness of the expected loss function $\bar{\ell}_x$, it holds that, for any $f, g \in L_2(P_X)$,

$$\mathcal{L}(f) + \langle g - f, \nabla_f \mathcal{L}(f) \rangle_{L_2} + \frac{L}{2} \|g - f\|_{L_2}^2$$

$$= \mathrm{E}_X \left[ \bar{\ell}_X(f(X)) + (g(X) - f(X))\bar{\ell}_X'(f(X)) + \frac{L}{2}(g(X) - f(X))^2 \right]$$

$$\geq \mathrm{E}_X \left[ \bar{\ell}_X(g(X)) \right] = \mathcal{L}(g),$$

where $\nabla_f \mathcal{L}$ is the Fréchet derivative in $L_2(P_X)$ (note that this inequality holds even though $\bar{\ell}_x(\cdot)$ is not a convex function). By substituting $g = f^*$ and $f = f^* + \frac{1}{L}\nabla_f \mathcal{L}(f_{h^*})$, we obtain

$$\mathcal{L}(f^*) + \frac{1}{2L}\|\nabla_f \mathcal{L}(f_{h^*})\|_{L_2}^2 \leq \mathcal{L}(f_{h^*})$$

$$\Rightarrow \quad \|\nabla_f \mathcal{L}(f_{h^*})\|_{L_2}^2 \leq 2L(\mathcal{L}(f_{h^*}) - \mathcal{L}(f^*)) \leq 2L\epsilon^2.$$

Therefore, for any $h \in \mathcal{H}$ such that $\|h - h^*\|_\mathcal{H} \leq \epsilon$, it holds that

$$\mathcal{L}(h) \leq \mathcal{L}(f_{h^*}) + \langle f_h - f_{h^*}, \nabla \mathcal{L}(f_{h^*}) \rangle_{L_2} + \frac{L}{2} \|f_h - f_{h^*}\|_{L_2}^2$$

$$\leq \mathcal{L}(f_{h^*}) + \frac{1}{2}\|f_h - f_{h^*}\|_{L_2}^2 + \frac{1}{2}\|\nabla \mathcal{L}(f_{h^*})\|_{L_2}^2 + \frac{L}{2}\|f_h - f_{h^*}\|_\infty^2$$

$$\leq \mathcal{L}(f_{h^*}) + \frac{1}{2}\|f_h - f_{h^*}\|_\infty^2 + \frac{1}{2}2L\epsilon^2 + \frac{L}{2}\|f_h - f_{h^*}\|_\infty^2$$

$$\leq \epsilon^2 + \frac{1+L}{2}(1+RD)^2 \|h - h^*\|_\mathcal{H}^2 + L\epsilon^2$$

$$\leq \left(1 + \frac{(1+L)(1+RD)^2}{2} + L\right)\epsilon^2 =: C_{(L,R,D)}\epsilon^2.$$

This yields that

$$-\log\tilde{\nu}_\beta(\{h \in \mathcal{H} : \mathcal{L}(h) - \mathcal{L}(f^*) \leq C_{(L,R,D)}\epsilon^2\})$$
$$\leq -\log\tilde{\nu}_\beta(\{h \in \mathcal{H} : \mathcal{L}(h) - \mathcal{L}(f^*) \leq C_{(L,R,D)}\epsilon^2, \|h - h^*\|_{\mathcal{H}} \leq \epsilon\})$$
$$\leq (1+\delta)\phi_{\beta,\lambda}(\epsilon).$$

Since $\delta$ is arbitrary, we obtain that

$$-\log\tilde{\nu}_\beta(\{h \in \mathcal{H} : \mathcal{L}(h) - \mathcal{L}(f^*) \leq C_{(L,R,D)}\epsilon^2\}) \leq \phi_{\beta,\lambda}(\epsilon). \tag{14}$$

By the Gaussian correlation inequality (Lemma 2), we have that

$$\phi_{\beta,\lambda}^{(0)}(\epsilon) = -\log\tilde{\nu}_\beta(\{W \in \mathcal{H} \mid W \in B_\epsilon\})$$
$$\leq -\log[\tilde{\nu}_\beta(\{h \in \mathcal{H} \mid \|h\|_{\mathcal{H}} \leq \epsilon\}) \times \tilde{\nu}_\beta(\{h \in \mathcal{H} \mid \|h\|_{\mathcal{H}_{\bar{K}\theta}} \leq \lambda_\beta^{-1/2}\bar{R}_\theta\})]$$
$$\leq -\log[\tilde{\nu}_\beta(\{h \in \mathcal{H} \mid \|h\|_{\mathcal{H}} \leq \epsilon\}) \times 1/2]$$
$$= -\log\tilde{\nu}_\beta(\{h \in \mathcal{H} \mid \|h\|_{\mathcal{H}} \leq \epsilon\}) + \log(2).$$

Therefore, we can see that

$$\inf_{h \in \mathcal{H}_{\bar{K}}:\mathcal{L}(h)-\mathcal{L}(f^*)\leq\epsilon^2/2} \lambda_\beta\|h\|_{\mathcal{H}_{\bar{K}}}^2 + \phi_{\beta,\lambda}^{(0)}(\epsilon) \leq \phi_{\beta,\lambda}(\epsilon).$$

Here we define $\epsilon^*$ as

$$\epsilon^* := \max\{\inf\{\epsilon > 0 : \phi_{\beta,\lambda}(\epsilon) \leq \beta\epsilon^2\}, n^{-\frac{1}{2(2-s)}}\}.$$

Note that, since $\phi_{\beta,\lambda}$ is monotonically non-increasing, $\epsilon^*$ satisfies

$$\phi_{\beta,\lambda}(\epsilon^*) \leq \beta\epsilon^{*2}. \tag{15}$$

Let $r > 1$, and $M_r := -2\Phi^{-1}(e^{-\beta\epsilon^{*2}r})$ where $\Phi$ is the cumulative distribution function of the standard normal distribution. Since $\Phi^{-1}(y) \geq -\sqrt{5/2\log(1/y)}$ for every $y \in (0, 1/2)$ and Eq. (15) implies $\log(2) \leq \beta\epsilon^{*2}$ yielding $e^{-\beta\epsilon^{*2}r} < e^{-\beta\epsilon^{*2}} \leq e^{-\log(2)} = 1/2$, $M_r$ is bounded by

$$-M_r/2 \geq -\sqrt{\frac{5}{2}\log(e^{\beta\epsilon^{*2}r})} = -\sqrt{\frac{5}{2}\beta\epsilon^{*2}r} \Rightarrow M_r \leq \sqrt{10\beta\epsilon^{*2}r}. \tag{16}$$

Let

$$\mathcal{F}_r := B_{\epsilon^*} + M_r\lambda_\beta^{-1/2}\mathcal{B}_{\mathcal{H}_{\bar{K}}}.$$

By Borell's inequality (Theorem 3.1 of [10]), the prior probability mass of $\mathcal{F}_r$ is lower bounded by

$$\tilde{\nu}_\beta(\mathcal{F}_r) \geq \Phi(M_r + \alpha_r)$$

where $\alpha_r \in \mathbb{R}$ is determined by

$$\alpha_r = \Phi^{-1}(\tilde{\nu}_\beta(B_{\epsilon^*})) = \Phi^{-1}(e^{-\phi_{\beta,\lambda}^{(0)}(\epsilon^*)}).$$

Since $\phi_{\beta,\lambda}^{(0)}(\epsilon^*) \leq \phi_{\beta,\lambda}(\epsilon^*) + \log(2) \leq \beta\epsilon^{*2} \leq \beta\epsilon^{*2}r$ (Eq. (15)), we have $\Phi(\alpha_r) \geq e^{-\beta\epsilon^{*2}r}$ by the definition of $\alpha_r$, which implies

$$\alpha_r \geq \Phi^{-1}(e^{-\beta\epsilon^{*2}r}) = -\frac{1}{2}M_r,$$

where the last equality is given by the definition of $M_r$. Therefore,

$$\tilde{\nu}_\beta(\mathcal{F}_r) \geq \Phi(M_r + \alpha_r) \geq \Phi(M_r/2) \geq 1 - \exp(-\beta\epsilon^{*2}r). \tag{17}$$

By the proof of Theorem 2.1 in [71], we obtain that the metric entropy of $\mathcal{F}_r$ is bounded by [2]

$$\log \mathcal{N}(\mathcal{F}_r, \|\cdot\|_{\mathcal{H}}, \epsilon^*) \leq \frac{1}{2} M_r^2 + \phi_{\beta,\lambda}^{(0)}(\epsilon^*),$$

and, more strongly, there exist $h_1, \ldots, h_{N_{\epsilon^*}} \in \mathcal{F}_r$ for $N_{\epsilon^*} := \mathcal{N}(\mathcal{F}_r, \epsilon^*, \|\cdot\|_{\mathcal{H}})$ such that

$$\mathcal{F}_r \subset B_{\epsilon^*} + \{h_1, \ldots, h_{N_{\epsilon^*}}\}.$$

This indicates that, even for smaller $\epsilon' \leq \epsilon^*$, it holds that

$$\log \mathcal{N}(\mathcal{F}_r, \|\cdot\|_{\mathcal{H}}, \epsilon') \leq N_{\epsilon^*} + \log \mathcal{N}(B_{\epsilon^*}, \|\cdot\|_{\mathcal{H}}, \epsilon').$$

Then, if we let $\tilde{\phi}(\epsilon') = \log \mathcal{N}((\lambda_\beta^{-1/2} \bar{R}_\theta) \mathcal{B}_{\mathcal{H}_{\tilde{K}^\theta}}, \|\cdot\|_{\mathcal{H}}, \epsilon')$, then for $B_{\epsilon^*} \subset (\lambda_\beta^{-1/2} \bar{R}_\theta) \mathcal{B}_{\mathcal{H}_{\tilde{K}^\theta}}$ by its definition, Eq. (15) and Eq. (16) give

$$\log \mathcal{N}(\mathcal{F}_r, \|\cdot\|_{\mathcal{H}}, \epsilon') \leq N_{\epsilon^*} + \tilde{\phi}(\epsilon') \leq \frac{1}{2} M_r^2 + \phi_{\beta,\lambda}^{(0)}(\epsilon^*) + \tilde{\phi}(\epsilon') \leq (5r+1)\beta\epsilon^{*2} + \tilde{\phi}(\epsilon'). \quad (18)$$

Note that if $\epsilon' > \epsilon^*$, then by using the fact $N_{\epsilon'} \leq N_{\epsilon^*}$, we can see that this inequality still holds: $\log \mathcal{N}(\mathcal{F}_r, \epsilon', \|\cdot\|_{\mathcal{H}}) = N_{\epsilon'} \leq N_{\epsilon^*} \leq N_{\epsilon^*} + \tilde{\phi}(\epsilon')$. The covering number of $\mathcal{B}_{\mathcal{H}_{\tilde{K}^\theta}}$ can be evaluated using the decay rate of the spectrum $(\mu_k(\tilde{K}^\theta))_k$ [64]. Indeed, $\mu_k(\tilde{K}^\theta) \lesssim k^{-\frac{\theta}{\tilde{\alpha}}}$ implies $\tilde{\phi}(\epsilon') \lesssim (\epsilon'/\lambda_\beta^{1/2})^{-2\tilde{\alpha}/\theta}$ [64, Theorem 15]. Moreover, the small ball probability $\phi_{\beta,\lambda}^{(0)}(\epsilon)$ can be evaluated using the covering number. First, notice that $\phi_{\beta,\lambda}^{(0)}(\epsilon) = -\log \tilde{\nu}_\beta(\{W \in \mathcal{H} \mid W \in B_\epsilon\}) \leq -\log \tilde{\nu}_\beta(\{W \in \mathcal{H} \mid \|W\|_{\mathcal{H}} \leq \epsilon\}) =: \varphi(\epsilon)$, and then [26] showed that

$$\varphi(2\epsilon) \lesssim \log \mathcal{N}\left(\lambda_\beta^{-1/2} \mathcal{B}_{\mathcal{H}_{\tilde{K}}}, \|\cdot\|_{\mathcal{H}}, \frac{\epsilon}{\sqrt{2\varphi(\epsilon)}}\right) \lesssim \varphi(\epsilon).$$

Here, since the entropy number in the middle is evaluated as $\log \mathcal{N}\left(\lambda_\beta^{-1/2} \mathcal{B}_{\mathcal{H}_{\tilde{K}}}, \|\cdot\|_{\mathcal{H}}, \epsilon\right) \lesssim (\epsilon/\lambda_\beta^{1/2})^{-2\tilde{\alpha}}$, we obtain

$$\phi_{\beta,\lambda}^{(0)}(\epsilon) \leq \varphi(\epsilon) \lesssim \left(\frac{\epsilon}{\lambda_\beta^{1/2}}\right)^{-\frac{2\tilde{\alpha}}{1-\tilde{\alpha}}}. \quad (19)$$

In this setting, we will show that, for any $r > 1$, there exists an event $\mathcal{E}_r$ with respect to data generation $D_n$ and exists $u^* > 0$ such that

A : $P(\mathcal{E}_r^c) \leq 2e^{-c' \min\{\beta\epsilon^{*2}, n\epsilon^{*2(2-s)}\}r}$ for a constant $c' > 0$,

B : $\widehat{\mathcal{L}}(W) - \widehat{\mathcal{L}}(f^*) \geq \frac{1}{2}[(\mathcal{L}(h) - \mathcal{L}(f^*)) - u^*r] \quad (\forall h \in \mathcal{F}_r)$

under the event $\mathcal{E}_r$,

C : $\mathrm{E}[\pi_\infty(\mathcal{F}_r^c)\mathbf{1}_{\mathcal{E}}] \leq 2\exp\left[-\frac{1}{2}(r-2)\beta\epsilon^{*2}\right]$,

D : $\mathrm{E}[\pi_\infty(\{W \in \mathcal{F}_r : \mathcal{L}(W) - \mathcal{L}(f^*) \geq 3ru^*\})\mathbf{1}_{\mathcal{E}_r}] \leq \exp\left[-\frac{1}{2}(r-2)\beta\epsilon^{*2}\right]$.

From now on, we will define $u^*$ and $\mathcal{E}_r$ and prove the conditions A, B, C, D one by one.

**Step 1: Definitions of $u^*$ and $\mathcal{E}_r$, and proof of A and B.**

For notational simplicity, we write $\ell(f, Z)$ to indicate $\ell(Y, f(X))$ for $Z = (X, Y)$. By Talangrand's concentration inequality [68, 12], we have

$$P\left(\sup_{h \in \mathcal{F}_r} \frac{|\mathcal{L}(h) - \mathcal{L}(f^*) - (\widehat{\mathcal{L}}(h) - \widehat{\mathcal{L}}(f^*))|}{\mathcal{L}(h) - \mathcal{L}(f^*) + u}\right.$$

$$\geq 2\mathrm{E}\left[\sup_{h\in\mathcal{F}_r}\frac{|\mathcal{L}(h)-\mathcal{L}(f^*)-(\widehat{\mathcal{L}}(h)-\widehat{\mathcal{L}}(f^*))|}{\mathcal{L}(h)-\mathcal{L}(f^*)+u}\right]+\sqrt{\frac{2t}{n}V}+\frac{2tU}{n}\right)$$

$$\leq\exp(-t),$$

for any $t\geq 1$, where

$$V=\sup_{h\in\mathcal{F}_r}\frac{\mathrm{E}[(\ell(h,Z)-\ell(f^*,Z)-\mathrm{E}[\ell(h,Z)-\ell(f^*,Z)])^2]}{(\mathcal{L}(h)-\mathcal{L}(f^*)+u)^2},$$

$$U=\sup_{h\in\mathcal{F}_r}\frac{\|\ell(h,\cdot)-\ell(f^*,\cdot)-\mathrm{E}[\ell(h,Z)-\ell(f^*,Z)]\|_\infty}{\mathcal{L}(h)-\mathcal{L}(f^*)+u}.$$

By the Bernstein condition, it holds that

$$\mathrm{Var}[\ell(f_h,Z)-\ell(f^*,Z)]\leq\mathrm{E}[(\ell(f_h,Z)-\ell(f^*,Z))^2]\leq C_B(\mathcal{L}(h)-\mathcal{L}(f^*))^s,$$

which gives

$$V\leq C_B\sup_{h\in\mathcal{F}_r}\frac{(\mathcal{L}(h)-\mathcal{L}(f^*))^s}{(\mathcal{L}(h)-\mathcal{L}(f^*)+u)^2}\leq\frac{C_B}{u^{2-s}}.$$

By the boundedness assumption of the loss function, we can see that

$$U\leq\frac{\bar{R}}{u}.$$

Hence, we have that

$$P\left(\sup_{h\in\mathcal{F}_r}\frac{|\mathcal{L}(h)-\mathcal{L}(f^*)-(\widehat{\mathcal{L}}(h)-\widehat{\mathcal{L}}(f^*))|}{\mathcal{L}(h)-\mathcal{L}(f^*)+u}\right.$$

$$\geq 2\mathrm{E}\left[\sup_{h\in\mathcal{F}_r}\frac{|\mathcal{L}(h)-\mathcal{L}(f^*)-(\widehat{\mathcal{L}}(h)-\widehat{\mathcal{L}}(f^*))|}{\mathcal{L}(h)-\mathcal{L}(f^*)+u}\right]+\sqrt{\frac{2C_B}{nu^{2-s}}t}+\frac{2\bar{R}}{nu}t\right)$$

$$\leq\exp(-t),\tag{20}$$

for any $t\geq 1$.

Hereafter, we bound the expectation of the supremum of the ratio type empirical process: $\mathrm{E}\left[\sup_{h\in\mathcal{F}_r}\frac{|\mathcal{L}(h)-\mathcal{L}(f^*)-(\widehat{\mathcal{L}}(h)-\widehat{\mathcal{L}}(f^*))|}{\mathcal{L}(h)-\mathcal{L}(f^*)+u}\right]$. Let the empirical $L_2$-norm be $\|h\|_n:=\sqrt{\frac{1}{n}\sum_{i=1}^n h(z_i)^2}$. By the usual Rademacher complexity and covering number argument (Lemma 11.4 of [11], Theorem 5.22 of [75] and Lemma A.5 of [6] for example), the non-ratio-type empirical process can be bounded as

$$\mathrm{E}\left[\sup_{h\in\mathcal{F}_r:\mathcal{L}(h)-\mathcal{L}(f^*)\leq u}|\mathcal{L}(h)-\mathcal{L}(f^*)-(\widehat{\mathcal{L}}(h)-\widehat{\mathcal{L}}(f^*))|\right]$$

$$\leq 2\mathrm{E}\left[\sup_{h\in\mathcal{F}_r:\mathcal{L}(h)-\mathcal{L}(f^*)\leq u}\left|\frac{1}{n}\sum_{i=1}^n\epsilon_i(\ell(f_h,z_i)-\ell(f^*,z_i)-(\mathcal{L}(h)-\mathcal{L}(f^*)))\right|\right]$$

$$\leq C\mathrm{E}\left[\inf_{a>0}\left\{a+\int_a^{\hat{r}(u)}\sqrt{\frac{\log\mathcal{N}(\{\ell(f_h,\cdot)-\ell(f^*,\cdot)\mid h\in\mathcal{F}_r,\ \mathcal{L}(h)-\mathcal{L}(f^*)\leq u\},\|\cdot\|_n,\epsilon')}{n}}\mathrm{d}\epsilon'\right\}\right],$$

where $\hat{r}(u):=\sup\{\|\ell(f_h,\cdot)-\ell(f^*,\cdot)\|_n\mid h\in\mathcal{F}_r,\ \mathcal{L}(h)-\mathcal{L}(f^*)\leq u\}$ and $C$ is a universal constant. The Dudley integral in the right hand side can be bounded by

$$\int_a^{\hat{r}(u)}\sqrt{\frac{\log\mathcal{N}(\{\ell(f_h,\cdot)-\ell(f^*,\cdot)\mid h\in\mathcal{F}_r,\ \mathcal{L}(h)-\mathcal{L}(f^*)\leq u\},\|\cdot\|_n,\epsilon')}{n}}\mathrm{d}\epsilon'$$

$$\overset{(1)}{\leq}\int_a^{\hat{r}(u)}\sqrt{\frac{\log\mathcal{N}(\{f_h\mid h\in\mathcal{F}_r\},\|\cdot\|_\infty,\epsilon'/B)}{n}}\mathrm{d}\epsilon'$$

$$\overset{(2)}{\leq}\int_a^{\hat{r}(u)}\sqrt{\frac{\log\mathcal{N}(\mathcal{F}_r,\|\cdot\|_\mathcal{H},\epsilon'/(B(1+RD)))}{n}}\mathrm{d}\epsilon'$$

$$\overset{(3)}{\leq} \int_a^{\hat{r}(u)} \sqrt{\frac{N_{\epsilon^*} + \tilde{\phi}(\epsilon'/(B(1+RD)))}{n}} d\epsilon',$$

where we used the bounded gradient condition on the loss function to show (1), used Lemma 1 to show (2), and used Eq. (18) to show (3). If we let $a = 1/n$, then we have

$$\mathrm{E}\left[\sup_{h \in \mathcal{F}_r : \mathcal{L}(h) - \mathcal{L}(f^*) \leq u} |\mathcal{L}(h) - \mathcal{L}(f^*) - (\widehat{\mathcal{L}}(h) - \widehat{\mathcal{L}}(f^*))|\right]$$

$$\leq C\mathrm{E}\left[\frac{1}{n} + \int_{1/n}^{\hat{r}(u)} \sqrt{\frac{N_{\epsilon^*} + \tilde{\phi}(\epsilon'/(B(1+RD)))}{n}} d\epsilon'\right] =: \psi_{r,\epsilon^*}(u).$$

Here, we assume that there exists an upper bound $\bar{\psi}_{r,\epsilon^*}(u)$ of $\psi_{r,\epsilon^*}(u)$ that satisfies

$$\bar{\psi}_{r,\epsilon^*}(4u) \leq 2\bar{\psi}_{r,\epsilon^*}(u) \quad (u > 0), \tag{21a}$$

$$\frac{\bar{\psi}_{r,\epsilon^*}(ur)}{ur} \leq \frac{\bar{\psi}_{1,\epsilon^*}(u)}{u} \quad (u > 0, \ r \geq 1). \tag{21b}$$

We will show these conditions in Step 5. Then, the so called *peeling device* gives

$$\mathrm{E}\left[\sup_{h \in \mathcal{F}_r} \frac{|\mathcal{L}(h) - \mathcal{L}(f^*) - (\widehat{\mathcal{L}}(h) - \widehat{\mathcal{L}}(f^*))|}{\mathcal{L}(h) - \mathcal{L}(f^*) + u}\right] \leq \frac{4\bar{\psi}_{r,\epsilon^*}(u)}{u}.$$

(Theorem 7.7 and Eq. (7.17) of [63]). Therefore, Eq. (20) can yields that

$$P\left(\sup_{h \in \mathcal{F}_r} \frac{|\mathcal{L}(h) - \mathcal{L}(f^*) - (\widehat{\mathcal{L}}(h) - \widehat{\mathcal{L}}(f^*))|}{\mathcal{L}(h) - \mathcal{L}(f^*) + u} \geq 8\frac{\bar{\psi}_{r,\epsilon^*}(u)}{u} + \sqrt{\frac{2C_B}{nu^{2-s}}t} + \frac{2\bar{R}}{nu}t\right) \leq \exp(-t).$$

Here, for $t_1 = \beta\epsilon^{*2}$, let $u^*$ be any real number satisfying

$$u^* \geq \max\left\{\epsilon^{*2}, \inf\left\{u > 0 : 8\frac{\bar{\psi}_{1,\epsilon^*}(u)}{u} + \sqrt{\frac{2C_B}{nu^{2-s}}t_1} + \frac{2\bar{R}}{nu}t_1 \leq \frac{1}{2}\right\}\right\}.$$

For more general $t = t_1 r = \beta\epsilon^{*2}r$, since $\frac{\bar{\psi}_{r,\epsilon^*}(u^*r)}{u^*r} \leq \frac{\bar{\psi}_{1,\epsilon^*}(u^*)}{u^*}$, combining with the fact that $\sqrt{\frac{2C_B}{n(u^*r)^{2-s}}t_1 r} \leq \sqrt{\frac{2C_B}{n(u^*)^{2-s}}t_1}$ and $\frac{2\bar{R}}{nu^*r}t_1 r = \frac{2\bar{R}}{nu^*}t_1$, it also holds that

$$8\frac{\bar{\psi}_{r,\epsilon^*}(u^*r)}{u^*r} + \sqrt{\frac{2C_B}{n(u^*r)^{2-s}}t_1 r} + \frac{2\bar{R}}{n(u^*r)}t_1 r \leq \frac{1}{2},$$

for $r \geq 1$. Therefore, the following inequality holds:

$$\widehat{\mathcal{L}}(h) - \widehat{\mathcal{L}}(f^*) \geq \frac{1}{2}[(\mathcal{L}(h) - \mathcal{L}(f^*)) - u^*r],$$

uniformly over all $h \in \mathcal{F}_r$ with probability $1 - e^{-\beta\epsilon^{*2}r}$. We denote this event by $\mathcal{E}_1$.

Eq. (14) gives that

$$-\log \tilde{\nu}_\beta(\{h \in \mathcal{H} : \mathcal{L}(h) - \mathcal{L}(f^*) \leq C_{(L,R,D)}\epsilon^{*2}\}) \leq \phi_{\beta,\lambda}(\epsilon^*).$$

Let the conditional probability measure of $\tilde{\nu}_\beta$ conditioned on the set $A_{\epsilon^*} := \{h \in \mathcal{H} : \mathcal{L}(h) - \mathcal{L}(f^*) \leq C_{(L,R,D)}\epsilon^{*2}\}$ be

$$\tilde{\nu}_\beta(B|A_{\epsilon^*}) := \frac{\tilde{\nu}_\beta(B \cap A_{\epsilon^*})}{\tilde{\nu}_\beta(A_{\epsilon^*})},$$

for a measurable set $B \subset \mathcal{H}$. Let $\bar{\ell}(Z) := \int \ell(h, Z)\tilde{\nu}_\beta(dh|A_{\epsilon^*})$. Then, we have that

$$\int \exp\left(-\beta(\widehat{\mathcal{L}}(h) - \widehat{\mathcal{L}}(f^*))\right) d\tilde{\nu}_\beta(h|A_{\epsilon^*}) \geq \exp\left(-\int \beta(\widehat{\mathcal{L}}(h) - \widehat{\mathcal{L}}(f^*))d\tilde{\nu}_\beta(h|A_{\epsilon^*})\right)$$

$$= \exp\left[-\beta\left(\frac{1}{n}\sum_{i=1}^{n}\bar{\ell}(z_i) - \mathrm{E}[\bar{\ell}(Z)]\right)\right].$$

Now, by the Bernstein's inequality,

$$P\left(\frac{1}{n}\sum_{i=1}^{n}\bar{\ell}(z_i) - \mathrm{E}[\bar{\ell}(Z)] \geq \sqrt{\frac{2C_B(C_{(L,R,D)}\epsilon^{*2})^s t}{n}} + \frac{\bar{R}t}{n}\right) \leq e^{-t},$$

for $t \geq 0$. Here, let $t = \frac{1}{8}\min\{\frac{1}{2C_B C_{(L,R,D)}^s}, \frac{1}{\bar{R}}\}n\min\{\epsilon^{*2(2-s)}, \epsilon^{*2}\}r$, then it holds that

$$P\left(\frac{1}{n}\sum_{i=1}^{n}\bar{\ell}(z_i) - \mathrm{E}[\bar{\ell}(Z)] \geq \frac{1}{2}\epsilon^{*2}r\right) \leq e^{-\frac{1}{8}\min\{\frac{1}{2C_B C_{(L,R,D)}^s}, \frac{1}{\bar{R}}\}n\min\{\epsilon^{*2(2-s)}, \epsilon^{*2}\}r}.$$

Therefore, this and the definition of $\tilde{\nu}_\beta(\cdot|A_{\epsilon^*})$ give that

$$\int \exp\left(-\beta(\widehat{\mathcal{L}}(h) - \widehat{\mathcal{L}}(f^*))\right) \mathrm{d}\tilde{\nu}_\beta(h) \geq \exp\left(-\tfrac{1}{2}\beta\epsilon^{*2}r\right)\tilde{\nu}_\beta(A_{\epsilon^*})$$

$$\geq \exp\left(-\tfrac{1}{2}\beta\epsilon^{*2}r - \beta\epsilon^{*2}\right) \quad (\because \text{Eqs. (14) and (15)})$$

$$\geq \exp\left(-(\tfrac{r}{2}+1)\beta\epsilon^{*2}\right). \tag{22}$$

with probability $1 - \exp[-\frac{1}{8}\min\{\frac{1}{2C_B C_{(L,R,D)}^s}, \frac{1}{\bar{R}}\}n\min\{\epsilon^{*2(2-s)}, \epsilon^{*2}\}r]$. We define this event as $\mathcal{E}_2$.

Combining $\mathcal{E}_1$ and $\mathcal{E}_2$, we define $\mathcal{E}_r = \mathcal{E}_1 \cap \mathcal{E}_2$, then $P(\mathcal{E}_r) \geq 1 - e^{-\beta\epsilon^{*2}r} - e^{-\frac{1}{8}\min\{\frac{1}{2C_B C_{(L,R,D)}^s}, \frac{1}{\bar{R}}\}n\min\{\epsilon^{*2(2-s)}, \epsilon^{*2}\}r} \geq 1 - 2e^{-c'\min\{\beta\epsilon^{*2}, n\epsilon^{*2(2-s)}\}r}$ for a constant $c' > 0$, where we used $\beta \leq n$.

**Step 2: Proof of** C.

Next, we evaluate the condition C. Eq. (22) gives that, on the event $\mathcal{E}_r$, the Bayes posterior probability of $\mathcal{F}_r^c$ is upper bounded by

$$\pi_\infty(\mathcal{F}_r^c) = \frac{\int_{h \in \mathcal{F}_r^c} \exp\left(-\beta(\widehat{\mathcal{L}}(h) - \widehat{\mathcal{L}}(f^*))\right) \mathrm{d}\tilde{\nu}_\beta(h)}{\int \exp\left(-\beta(\widehat{\mathcal{L}}(h) - \widehat{\mathcal{L}}(f^*))\right) \mathrm{d}\tilde{\nu}_\beta(h)}$$

$$\leq \exp\left((\tfrac{r}{2}+1)\beta\epsilon^{*2}\right)\int_{h \in \mathcal{F}_r^c} \exp\left(-\beta(\widehat{\mathcal{L}}(h) - \widehat{\mathcal{L}}(f^*))\right) \mathrm{d}\tilde{\nu}_\beta(h).$$

Therefore, it holds that

$$\mathrm{E}[\pi_\infty(\mathcal{F}_r^c)\mathbf{1}_{\mathcal{E}_r}] \leq \mathrm{E}\left[\mathbf{1}_{\mathcal{E}_r}\exp\left((\tfrac{r}{2}+1)\beta\epsilon^{*2}\right)\int_{h \in \mathcal{F}_r^c} \exp(-\beta(\widehat{\mathcal{L}}(h) - \widehat{\mathcal{L}}(f^*)))\mathrm{d}\tilde{\nu}_\beta(h)\right]$$

$$\leq \exp\left((\tfrac{r}{2}+1)\beta\epsilon^{*2}\right)\int_{h \in \mathcal{F}_r^c} \mathrm{E}\left[\exp\left(-n\frac{\beta}{n}(\widehat{\mathcal{L}}(h) - \widehat{\mathcal{L}}(f^*))\right)\right]\mathrm{d}\tilde{\nu}_\beta(h)$$

$$= \exp\left((\tfrac{r}{2}+1)\beta\epsilon^{*2}\right)\int_{h \in \mathcal{F}_r^c} \prod_{i=1}^{n} \mathrm{E}_{Z_i}\left[\exp\left(-\frac{\beta}{n}(\ell(h, Z_i) - \ell(f^*, Z_i))\right)\right]\mathrm{d}\tilde{\nu}_\beta(h)$$

$$\leq \exp\left((\tfrac{r}{2}+1)\beta\epsilon^{*2}\right)\tilde{\nu}_\beta(\mathcal{F}_r^c) \quad (\because \text{predictor condition of Assumption 2})$$

$$\leq 2\exp\left((\tfrac{r}{2}+1)\beta\epsilon^{*2} - \beta\epsilon^{*2}r\right) \quad (\because \text{Eq. (17)})$$

$$= 2\exp\left(-\tfrac{1}{2}(r-2)\beta\epsilon^{*2}\right).$$

**Step 3: Proof of** D.

Next, we prove the condition D. Similarly to C, we have that

$$
\mathrm{E}[\pi_\infty(\{W \in \mathcal{F}_r : \mathcal{L}(W) - \mathcal{L}(f^*) \geq 3ru^*\})\mathbf{1}_{\mathcal{E}_r}]
$$

$$
= \mathrm{E}\left[ \frac{\int_{\mathcal{L}(h)-\mathcal{L}(f^*)\geq 3ru^*} \exp(-\beta(\widehat{\mathcal{L}}(h) - \widehat{\mathcal{L}}(f^*)))\mathrm{d}\tilde{\nu}_\beta(h)}{\int \exp(-\beta(\widehat{\mathcal{L}}(h) - \widehat{\mathcal{L}}(f^*)))\mathrm{d}\tilde{\nu}_\beta(h)} \mathbf{1}_{\mathcal{E}_r} \right]
$$

$$
\leq \mathrm{E}\left[ \mathbf{1}_{\mathcal{E}_r} \exp\left(\left(\tfrac{r}{2}+1\right)\beta\epsilon^{*2}\right) \int_{\mathcal{L}(h)-\mathcal{L}(f^*)\geq 3ru^*} \exp(-\beta(\widehat{\mathcal{L}}(h) - \widehat{\mathcal{L}}(f^*)))\mathrm{d}\tilde{\nu}_\beta(h) \right]
$$

$$
\leq \mathrm{E}\left[ \mathbf{1}_{\mathcal{E}_r} \exp\left(\left(\tfrac{r}{2}+1\right)\beta\epsilon^{*2}\right) \int_{\widehat{\mathcal{L}}(h)-\widehat{\mathcal{L}}(f^*)\geq ru^*} \exp(-\beta(\widehat{\mathcal{L}}(h) - \widehat{\mathcal{L}}(f^*)))\mathrm{d}\tilde{\nu}_\beta(h) \right]
$$

$$
(\because \text{condition B is satisfied on } \mathcal{E}_r)
$$

$$
\leq \exp\left(\left(\tfrac{r}{2}+1\right)\beta\epsilon^{*2} - r\beta u^*\right)
$$

$$
\leq \exp\left[-\tfrac{1}{2}(r-2)\beta\epsilon^{*2}\right].
$$

**Step 4: Integrating all bounds of** A, B, C, D**.**

Finally, we integrate all bounds to obtain an excess risk bound.

$$
\mathrm{E}_{D^n}\left[ \int \mathcal{L}(W) - \mathcal{L}(f^*)\mathrm{d}\pi_\infty(W) \right]
$$

$$
= \mathrm{E}_{D^n}\left[ \int_0^\infty \pi_\infty(\{W \in \mathcal{H} : \mathcal{L}(W) - \mathcal{L}(f^*) > t\})\mathrm{d}t \right]
$$

$$
= \int_0^\infty \mathrm{E}_{D^n}\left[ \pi_\infty(\{W \in \mathcal{H} : \mathcal{L}(W) - \mathcal{L}(f^*) > t\}) \right]\mathrm{d}t \quad \text{(by Fubuni's theorem)}
$$

$$
= 3u^* + \int_1^\infty 3u^* \mathrm{E}_{D^n}\left[ \pi_\infty(\{W \in \mathcal{H} : \mathcal{L}(W) - \mathcal{L}(f^*) > 3ru^*\}) \right]\mathrm{d}r
$$

$$
\leq 3u^* + 3u^* \int_1^\infty \mathrm{E}_{D^n}\left\{ \mathbf{1}_{\mathcal{E}_r^c} + \mathbf{1}_{\mathcal{E}_r}[\pi_\infty(\mathcal{F}_r^c) + \pi_\infty(\{W \in \mathcal{F}_r : \mathcal{L}(W) - \mathcal{L}(f^*) > 3ru^*\})] \right\}\mathrm{d}r
$$

$$
\leq 3u^* + 3u^* \int_1^\infty \min\left\{ 2e^{-c'\min\{\beta\epsilon^{*2}, n\epsilon^{*2(2-s)}\}r} + 3\exp\left(-\tfrac{1}{2}(r-2)\beta\epsilon^{*2}\right), 1 \right\}\mathrm{d}r
$$

$$
\lesssim u^*, \tag{23}
$$

where in the last inequality, we used $\beta\epsilon^{*2} \geq \log(2)$ by Eq. (15) and $n\epsilon^{*2(2-s)} \geq 1$ by the definition of $\epsilon^*$.

**Step 5: Evaluation of** $u^*$**.**

Based on the arguments above, our goal is reduced to evaluating $u^*$. We note that

$$
\mathrm{E}[\hat{r}(u)^2] \leq u^s + \bar{R}\psi_{r,\epsilon^*}(u) \leq u^s + \bar{R}\mathrm{E}\left[ \int_0^{\hat{r}(u)} \sqrt{\frac{\beta\epsilon^{*2}r + \left(\epsilon'/\lambda_\beta^{1/2}(B(1+RD))^{-1}\right)^{-2\tilde{\alpha}/\theta}}{n}}\mathrm{d}\epsilon' \right]
$$

$$
\lesssim u^s + \sqrt{\frac{\beta\epsilon^{*2}r}{n}\mathrm{E}[\hat{r}^2(u)]} + \frac{1}{\sqrt{n}}\lambda_\beta^{\tilde{\alpha}/\theta}\mathrm{E}[(\hat{r}(u))^{1-\tilde{\alpha}/\theta}]
$$

$$
\leq u^s + \sqrt{\frac{\beta\epsilon^{*2}r}{n}\mathrm{E}[\hat{r}^2(u)]} + \frac{1}{\sqrt{n}}\lambda_\beta^{\tilde{\alpha}/\theta}(\mathrm{E}[\hat{r}(u)^2])^{(1-\tilde{\alpha}/\theta)/2}.
$$

Therefore, we have that

$$
\mathrm{E}[\hat{r}(u)^2] \lesssim u^s \vee \frac{\beta}{n}\epsilon^{*2}r \vee n^{-\frac{1}{1+\tilde{\alpha}/\theta}}\lambda_\beta^{\frac{2\tilde{\alpha}/\theta}{1+\tilde{\alpha}/\theta}}.
$$

This gives that

$$\psi_{r,\epsilon^*}(u) \lesssim \mathrm{E}\left[\int_0^{\hat{r}(u)} \sqrt{\frac{\beta\epsilon^{*2}r + \left(\epsilon'/\lambda_\beta^{1/2}(B(1+RD))^{-1}\right)^{-2\tilde{\alpha}/\theta}}{n}}\,\mathrm{d}\epsilon'\right]$$

$$\lesssim \sqrt{\frac{\beta\epsilon^{*2}r}{n}\mathrm{E}[\hat{r}(u)^2]} + \frac{1}{\sqrt{n}}\lambda_\beta^{\tilde{\alpha}/\theta}(\mathrm{E}[\hat{r}(u)^2])^{(1-\tilde{\alpha}/\theta)/2}$$

$$\lesssim \frac{\beta}{n}\epsilon^{*2}r + n^{-\frac{1}{1+\tilde{\alpha}/\theta}}\lambda_\beta^{\frac{2\tilde{\alpha}/\theta}{1+\tilde{\alpha}/\theta}} + u^{s/2}\sqrt{\frac{\beta\epsilon^{*2}r}{n} + n^{-\frac{1}{1+\tilde{\alpha}/\theta}}\lambda_\beta^{\frac{2\tilde{\alpha}/\theta}{1+\tilde{\alpha}/\theta}}} \quad (\because \text{Young's inequality}).$$

Here, we let the upper bound in the right hand side as $\bar{\psi}_{r,\epsilon^*}$, then we can easily show the condition (21), that is, $\bar{\psi}_{r,\epsilon^*}(4u) \leq 2\bar{\psi}_{r,\epsilon^*}(u)$ and $\frac{\bar{\psi}_{r,\epsilon^*}(ur)}{ur} \leq \frac{\bar{\psi}_{1,\epsilon^*}(u)}{u}$. Finally, by the definition of $u^*$, we obtain that

$$u^* \lesssim \epsilon^{*2} \vee \left(\frac{\beta}{n}\epsilon^{*2} + n^{-\frac{1}{1+\tilde{\alpha}/\theta}}\lambda_\beta^{\frac{2\tilde{\alpha}/\theta}{1+\tilde{\alpha}/\theta}}\right)^{\frac{1}{2-s}} \vee \frac{1}{n}.$$

This yields the assertion. $\qquad\qquad\qquad\qquad\qquad\qquad\qquad\qquad\qquad\qquad\qquad\qquad\square$

### D.3 Proof of fast rate for classification (Theorem 3)

*Proof.* Let the convergence rate in the right hand side of Eq. (9) in Theorem 2 be $u^*$.

Since both $\bar{W}_2(a)$ and $\bar{W}_1(w)$ are bounded and the activation function $\sigma$ is included in the Hölder class $\mathcal{C}^m(\mathbb{R})$, the model $\{f_W \mid W \in \mathcal{H}\}$ is also included in the Hölder class $\mathcal{C}^m(\mathcal{X})$ with regularity $m$ and especially it is included in the Sobolev space $W_2^m(\mathcal{X})$:

$$f_W \in W_2^m(\mathcal{X}).$$

Moreover, since the logistic loss is $C^\infty$-class and its derivative up to $m$-th order is upper bounded, the function $x \mapsto \ell(f_W(x), y)$ is also included in $W^m(\mathcal{X})$ for all $y \in \{\pm 1\}$. Therefore, $\hat{h}_W(x) := \mathrm{E}_{Y|x}[\ell(f_W(x), Y)](= h(f_W(x)|x))$ is also included in $W^m(\mathcal{X})$. Moreover, $\|\hat{h}_W\|_{W_2^m(\mathcal{X})} \leq C$ uniformly over all $W \in \mathcal{H}$.

If $X_0$ and $X_1$ are a pair of quasi-normed spaces which are continuously embedded in a linear Hausdorff space $\mathcal{G}$, their $K$-functional is defined for any $f \in X_0 + X_1$ by

$$K(f, t; X_0, X_1) := \inf_{f=f_0+f_1} \|f_0\|_{X_0} + \|f_1\|_{X_1}.$$

For each $0 < \theta < 1$, $0 < p \leq \infty$, the *interpolation space* $[X_0, X_1]_{q,\theta}$ is the set of all functions $f \in X_0 + X_1$ for which

$$\|f\|_{[X_0,X_1]_{q,\theta}} := \left(\int_0^\infty (t^{-\theta}K(f, t; X_0, X_1))^q \frac{\mathrm{d}t}{t}\right)^{1/q}$$

is finite. For $q = \infty$, the right hand side is properly modified in a usual manner. As shown by [21], it holds that

$$[L_2(\Omega), W_2^m(\mathcal{X})]_{1,d/2m} = B_{2,1}^{d/2}(\mathcal{X}),$$

where $L_2(\mathcal{X})$ is the $L_2$-space on $\mathcal{X}$ with respect to the Lebesgue measure and $B_{2,1}^{d/2}(\mathcal{X})$ is the Besov space defined on $\mathcal{X}$ (see [21] for its definition). Note that $d/2m < 1$ by the assumption. From this property, combined the extension theorem of [21] and the embedding property of the Besov space [69], we have that $B_{2,1}^{d/2}(\mathcal{X}) \hookrightarrow L_\infty(\mathcal{X})$. Under this condition, it is known that the following inequality holds

$$\|\hat{h}_W - h^*\|_\infty \leq C\|\hat{h}_W - h^*\|_{L_2(\mathcal{X})}^{1-\frac{d}{2m}}\|\hat{h}_W - h^*\|_{W_2^m(\mathcal{X})}^{\frac{d}{2m}}$$

$$\leq Cc_0^{-(1-d/2m)}\|\hat{h}_W - h^*\|_{L_2(P_X)}^{1-\frac{d}{2m}}\|\hat{h}_W - h^*\|_{W_2^m(\mathcal{X})}^{\frac{d}{2m}},$$

(see [9, 64]). Combining this with the assumption $h^* \in W_2^m(\mathcal{X})$ and the fact $\|\hat{h}_W\|_{W_2^m(\mathcal{X})} \le C$, if $\|\hat{h}_W - h^*\|_{L_2(P_X)} \le \epsilon$ for sufficiently small $\epsilon$, we have an $L_\infty$-norm bound as $\|\hat{h}_W - h^*\|_\infty \le C'\epsilon^{1-\frac{d}{2m}}$. Thus, if we choose $\epsilon$ so that $\epsilon^{1-\frac{d}{2m}} = \Theta(\delta)$ and let $W$ satisfy $\|\hat{h}_W - h^*\|_{L_2(P_X)} \le \epsilon$, then we can have

$$\|\hat{h}_W - h^*\|_\infty < \delta/2.$$

Then, by the assumption that $h^*(x) \le \log(2) - \delta$, it holds that

$$\hat{h}_W(x) < \log(2) - \delta/2 \quad \text{(a.s.)},$$

which indicates that

$$P_X(\text{sign}(f_W(X)) = g^*(X)) = 1.$$

Therefore, we only need to bound the quantity $\|\hat{h}_{W_k} - h^*\|_{L_2(P_X)}^2$ for $W_k \sim \pi_k$.

Here, we show that the Bernstein condition (Assumption 2) is satisfied with $s = 1$ under Assumptions 3. By Assumptions 3 and $\|f_W\|_\infty \le R$ for any $W \in \mathcal{H}$ by the definition of the clipping operator, it holds that $\|f^*\|_\infty \le R$. Therefore, Lemma 3 yields the Bernstein condition with $s = 1$ and $C_B = 4 + 3R$. Therefore, $\|\hat{h}_W - h^*\|_{L_2(P_X)}^2$ can be bounded as

$$\|\hat{h}_W - h^*\|_{L_2(P_X)}^2$$
$$\le \text{E}_Z[(\ell(f_W, Z) - \ell(f^*, Z))^2]$$
$$\le C_B(\mathcal{L}(W) - \mathcal{L}(f^*))^s = C_B(\mathcal{L}(W) - \mathcal{L}(f^*)),$$

where we used Jensen's inequality in the first inequality, and we applied $s = 1$ in the last equality. First, we consider the stationary distribution. For any $\epsilon' > u^*$, we have already shown in the proof of Theorem 2 (See Eq. (23)) that

$$\text{E}_{D^n}[\pi_\infty(\{W \in \mathcal{H} \mid \mathcal{L}(W) - \mathcal{L}(f^*) \ge \epsilon'\})]$$
$$\le C \exp(-c\beta u^* \times (\epsilon'/u^*)) = C \exp(-c\beta\epsilon'). \tag{24}$$

Next, we consider the intermediate solution $W_k$. Suppose that the sample size $n$ is sufficiently large and $\lambda$ is appropriately chosen with sufficiently large $\beta$ so that $u^* \ll \delta^{2m/(2m-d)3}$. The probability of misclassification is bounded by

$$\text{E}[\pi_k(\{W_k \in \mathcal{H} \mid P_X(\text{sign}(f_{W_k}(X)) = \text{sign}(f^*(X))) \ne 1\})]$$
$$\le \text{E}[P_{W_k \sim \pi_k}[\mathcal{L}(W_k) - \mathcal{L}(f^*) \ge \epsilon/C_B]]$$
$$= \text{E}[P_{W_k \sim \pi_k, W \sim \pi_\infty}[\mathcal{L}(W_k) - \mathcal{L}(W) - (\mathcal{L}(f^*) - \mathcal{L}(W)) \ge \epsilon/C_B]]$$
$$\le \text{E}[P_{W_k \sim \pi_k, W \sim \pi_\infty}[\mathcal{L}(W_k) - \mathcal{L}(W) \ge \epsilon/(2C_B)]] + \text{E}[P_{W \sim \pi_\infty}[\mathcal{L}(W) - \mathcal{L}(f^*) \ge \epsilon/(2C_B)]]$$
$$\le \text{E}[\text{E}_{W_k \sim \pi_k, W \sim \pi_\infty}[\mathcal{L}(W_k) - \mathcal{L}(W)]/(\epsilon/(2C_B))] + \text{E}[P_{W \sim \pi_\infty}[\mathcal{L}(W) - \mathcal{L}(f^*) \ge \epsilon/(2C_B)]]$$
$$\lesssim \frac{\Xi_k}{\delta^{2m/(2m-d)}} + \exp(-c'\beta\delta^{2m/(2m-d)}),$$

where we used $\epsilon = \Theta(\delta^{2m/(2m-d)})$ and Eq. (24) in the last inequality. Therefore, for a fixed $\delta$, we can obtain the Bayes classifier with high probability by setting $\eta$ sufficiently small and taking sufficiently large $k$.

**Making the first term as large as the second term.**

We see that the first term in the right hand side is coming from the bound of $\text{E}[P_{W_k \sim \pi_k, W \sim \pi_\infty}[\mathcal{L}(W_k) - \mathcal{L}(W) \ge \epsilon/(2C_B)]]$. To bound this, we used the following bound:

$$P_{W_k \sim \pi_k, W \sim \pi_\infty}[\mathcal{L}(W_k) - \mathcal{L}(W) \ge \epsilon/(2C_B)] \le \text{E}_{W_k \sim \pi_k, W \sim \pi_\infty}[\mathcal{L}(W_k) - \mathcal{L}(W)]/(\epsilon/(2C_B))$$
$$\lesssim \frac{\Xi_k}{\delta^{2m/(2m-d)}},$$

almost surely. Therefore, if $\Xi_k$ is sufficiently small such that $\frac{\Xi_k}{\delta^{2m/(2m-d)}} \ll 1$, then we have

$$P_{W_k \sim \pi_k, W \sim \pi_\infty}[\mathcal{L}(W_k) - \mathcal{L}(W) \geq \epsilon/(2C_B)] \leq 1/2.$$

Therefore, by running the algorithm $S$-times and picking up the beset $W_k$ in terms of the validation error (write it as $W_k^{(S)}$), then we have that

$$P_{W_k \sim \pi_k, W \sim \pi_\infty}[\mathcal{L}(W_k^{(S)}) - \mathcal{L}(W) \geq \epsilon/(2C_B)] \leq 1/2^S.$$

Thus, for sufficiently large $S$ such that the right hand side can be smaller than the second term $\exp(-c'\beta\delta^{2m/(2m-d)})$, we have that

$$\mathrm{E}\left\{ P_{W_k^{(S)}}\left[ P_X\left(\mathrm{sign}(f_{W_k^{(S)}}(X)) = \mathrm{sign}(f^*(X))) \neq 1 \mid D^n \right] \right\} \lesssim \exp(-c'\beta\delta^{2m/(2m-d)}).$$

$\square$

**Lemma 3.** *Suppose that $\|f^*\|_\infty \leq R$ and $\sup_W \|f_W\|_\infty \leq R$. Then, the logistic loss satisfies the Bernstein condition with $s = 1$ and $C_B = 4 + 3R$.*

*Proof.* Since $\|f^*\|_\infty \leq R$, it holds that $|h^*(x)| = |h(f^*(x)|x)|$ is also bounded by $R$ (a.s.). Here, we fix $x \in \mathcal{X}$ and write $p = P(Y = 1|X = x)$. By the optimality of $f^*(x)$, we have that $p = \frac{1}{1+\exp(-f^*(x))}$. Accordingly, we denote $q = \frac{1}{1+\exp(-f_W(x))}$ for any $W \in \mathcal{H}$.

Then, what we need to show is that

$$p\left[\log\left(\frac{p}{q}\right)\right]^2 + (1-p)\left[\log\left(\frac{1-p}{1-q}\right)\right]^2 \leq C_B\left\{ p\log\left(\frac{p}{q}\right) + (1-p)\log\left(\frac{1-p}{1-q}\right) \right\}. \quad (25)$$

The right hand side can be rewritten as

$$p\left[\log\left(\frac{p}{q}\right) + \frac{1}{p}(q-p)\right] + (1-p)\left[\log\left(\frac{1-p}{1-q}\right) - \frac{1}{1-p}(q-p)\right],$$

and by noticing the convexity of $-\log(\cdot)$, each term of the right hand side is non-negative. We show the inequality (25) by showing

$$\left[\log\left(\frac{p}{q}\right)\right]^2 \leq C_B\left[\log\left(\frac{p}{q}\right) + \frac{1}{p}(q-p)\right], \quad (26)$$

$$\left[\log\left(\frac{1-p}{1-q}\right)\right]^2 \leq C_B\left[\log\left(\frac{1-p}{1-q}\right) - \frac{1}{1-p}(q-p)\right]. \quad (27)$$

Without loss of generality, we may assume $p \leq 1/2$.

**Step 1: Proof of Eq.** (26). We show the inequality by considering the following four settings (i) $p/2 \leq q \leq p$, (ii) $q < p/2$, (iii) $p \leq q \leq 2p$, (iv) $2p < q$. Let $f_1(q) = \log(p/q) + \frac{1}{p}(q-p)$ and $f_2(q) = [\log(p/q)]^2$.

(i) $(p/2 \leq q \leq p)$ Since $f_1(q)$ is a convex function satisfying $\frac{\mathrm{d}^2}{\mathrm{d}q^2}f_1(q) = 1/q^2 \geq 1/p^2$ $(\forall q \leq p)$, $f_1(p) = 0$ and $f_1(q) \geq 0$, it holds that $f_1(q) \geq \frac{1}{p^2}(q-p)^2$ for all $q \leq p$. On the other hand, $0 \leq \log(p/q) \leq \frac{2}{p}(p-q)$ for $p/2 \leq q \leq p$, it holds that $f_2(q) \leq \frac{4}{p^2}(q-p)^2$ $(p/2 \leq \forall q \leq p)$. These inequalities yield

$$4f_1(q) \geq f_2(q) \quad (p/2 \leq \forall q \leq p).$$

(ii) $(q < p/2)$. Since $f_1(q) \geq \frac{1}{p^2}(p-q)^2$ $(\forall q \leq p)$ and $-\frac{1}{p}(q-p) \leq 2\frac{1}{p^2}(q-p)^2 \leq 2f_1(q)$ $(\forall q \leq p/2)$, we have that

$$-\frac{1}{3}\log(p/q) \leq \frac{1}{p}(q-p) \leq 0 \quad (\forall q \leq p/2).$$

Therefore, by the definition of $f_1$, we have

$$f_1(q) \geq \frac{2}{3}\log(p/q) \geq \frac{2}{3\log(p/q)}[\log(p/q)]^2 \geq \frac{2}{3\log(1+\exp(R))}f_2(q),$$

where we used $p \leq 1$ and $q = \frac{1}{1+\exp(-f_W(x))} \geq \frac{1}{1+\exp(R)}$.

(iii) ($p \leq q \leq 2p$). In this setting, the convexity of $-\log(\cdot)$ gives $0 \geq \log(p/q) = -\log(q/p) \geq \frac{1}{p}(p-q)$. Therefore, it holds that $f_2(q) \leq \frac{1}{p^2}(p-q)^2$. On the other hand, since $f_2''(q) = \frac{1}{q^2} \geq \frac{1}{4p^2}$, it holds that $f_1(x) \geq \frac{1}{4p^2}(p-q)^2$. Therefore, we have

$$4f_1(q) \geq f_2(q) \quad (p \leq \forall q \leq 2p).$$

(iv) ($2p < q$). By the convexity of $-\log(q)$, we have that

$$-\frac{1}{\log(2)}\log(q/p) \geq \frac{1}{\log(2)}[-\log(q)+\log(p)] \geq \frac{1}{\log(2)}[-\log(2p) - \frac{1}{2p}(q-2p)+\log(p)]$$

$$= \frac{1}{\log(2)}[-\log(2) - \frac{1}{2p}(q-2p)] = -1 - \frac{1}{2\log(2)p}(q-2p)$$

$$\geq -\frac{1}{p}(q-2p) - 1 = -\frac{1}{p}(q-p) \quad (\forall q > 2p),$$

where we used $2\log(2) \geq 1$. This yields that

$$f_1(q) \geq \left(1 - \frac{1}{\log(2)}\right)\log(p/q) = \frac{1-\log(2)}{\log(2)}\log(q/p) \geq 0 \quad (\forall q > 2p).$$

(Remember that $\log(2) < 1$). Therefore,

$$f_1(q) \geq \frac{1-\log(2)}{\log(2)\log(q/p)}[\log(q/p)]^2 \geq \frac{1-\log(2)}{\log(2)\log(1+\exp(R))}[\log(q/p)]^2$$

$$= \frac{1-\log(2)}{\log(2)\log(1+\exp(R))}f_2(q) \quad (\forall q > 2p),$$

where we used $q \leq 1$ and $p \geq \frac{1}{1+\exp(R)}$.

**Step 2: Proof of Eq.** (27). This is shown completely in the same manner with the proof of Eq. (26) by setting $p \leftarrow 1-p$ and $q \leftarrow 1-q$.

**Step 3.** Combining the results of Step 1 and Step 3, we have the equations (26) and (27) with

$$C_B = \max\left\{4, \frac{3}{2}\log(1+\exp(R)), \frac{\log(2)}{1-\log(2)}\log(1+\exp(R))\right\} \leq 4 + 3R,$$

where we used $\frac{3}{2} \leq \frac{\log(2)}{1-\log(2)} \leq 3$, $\log(1+\exp(R)) \leq \log(2) + R$ by the Lipschitz continuity of $x \mapsto \log(1+\exp(x))$, and $\frac{\log^2(2)}{1-\log(2)} \leq 2$. Therefore, by resetting $C_B = 4 + 3R$, we obtain the assertion. $\square$

### D.4 Derivation of the fast rate of regression (Eq. (10))

Since $f^*$ is realized by $f_{W^*}$, $\|f_W\|_\infty \leq R$ for any $W \in \mathcal{H}$ and $|\epsilon_i| \leq C$, we have that

$$\ell(Y, f_W(X)) = (Y - f_W(X))^2 = (f_{W^*} + \epsilon - f_W(X))^2 \leq 2[(f_{W^*}(X) - f_W(X))^2 + \epsilon^2] \leq 2(4R^2 + C^2).$$

Therefore, the assumption $0 \leq \ell(Y, f_W(X)) \leq \bar{R}$ ($\forall W \in \mathcal{H}$) (a.s.) is obtained by $\bar{R} = 2(4R^2 + C^2)$. Other assumptions in

Write $\mathcal{H}_{\tilde{K}^\theta} := \mathcal{H}_{K^{\theta(\gamma+1)}}$. As we have stated in the main text, we can show the "bias" and "variance" terms can be bounded as

$$\inf_{h \in \mathcal{H}_{\tilde{K}} : \mathcal{L}(h) - \mathcal{L}(f^*) \leq \epsilon^2} \lambda_\beta \|h\|_{\mathcal{H}_{\tilde{K}}}^2 \lesssim \lambda_\beta \epsilon^{-\frac{2(1-\theta)}{\theta}},$$

$$-\log \tilde{\nu}_\beta(\{h \in \mathcal{H} : \|h\|_{\mathcal{H}} \leq \epsilon\}) \lesssim (\epsilon/\lambda_\beta^{1/2})^{-\frac{2\bar{\alpha}}{1-\bar{\alpha}}}.$$

The variance term has been already evaluated in Eq. (19). Now, we evaluate the bias term. By the definitions of $\mathcal{H}_{\tilde{K}}$, $W^* \in \mathcal{H}_{\tilde{K}}$ means that there exists $(a_k)_{k=0}^\infty$ such that

$$W^* = \sum_{k=0}^\infty \sqrt{\mu_k^{\theta(\gamma+1)}} a_k e_k, \text{ and } \sum_{k=0}^\infty a_k^2 < \infty.$$

Here, we denote $Q := \sum_{k=0}^{\infty} a_k^2$. Now, let $\tilde{W} = \sum_{k=0}^{N} \sqrt{\mu_k^{\theta(\gamma+1)}} a_k e_k$ for some $N \in \mathbb{N}$ as an approximator of $W^*$. Then, its norm in $\mathcal{H}_{\tilde{K}}$ can be evaluated as

$$\|\tilde{W}\|_{\mathcal{H}_{\tilde{K}}}^2 = \sum_{k=0}^{N} \mu_k^{-(\gamma+1)} \mu_k^{\theta(\gamma+1)} a_k^2 = \sum_{k=0}^{N} \mu_k^{(\theta-1)(\gamma+1)} a_k^2. \tag{28}$$

We evaluate the discrepancy between $W^*$ and $\tilde{W}$ and evaluate its norm in $\mathcal{H}$. Since $W^* - \tilde{W} = \sum_{k=N+1}^{\infty} \sqrt{\mu_k^{\theta(\gamma+1)}} a_k e_k$, its $\mathcal{H}$-norm is given by

$$\|W^* - \tilde{W}\|_{\mathcal{H}}^2 = \sum_{k=N+1}^{\infty} \mu_k^{\theta(\gamma+1)} a_k^2. \tag{29}$$

Note that $\mathcal{L}(f_{\tilde{W}}) - \mathcal{L}(f^*) = \|f_{\tilde{W}} - f^*\|_{L_2(P_X)}^2 \le (1+RD)^2 \|\tilde{W} - W^*\|_{\mathcal{H}}^2$ by Lemma 1. Therefore, to ensure $\mathcal{L}(f_{\tilde{W}}) - \mathcal{L}(f^*) \le \epsilon^2$, it suffices to let $(1 + RD)^2 \|\tilde{W} - W^*\|_{\mathcal{H}}^2 \le \epsilon^2$. By Eq. (29), this means that $\sum_{k=N+1}^{\infty} \mu_k^{\theta(\gamma+1)} a_k^2 \le \epsilon^2/(1 + RD)^2$. Here, note that

$$\|W^* - \tilde{W}\|_{\mathcal{H}}^2 = \sum_{k=N+1}^{\infty} \mu_k^{\theta(\gamma+1)} a_k^2 \le \mu_{N+1}^{\theta(\gamma+1)} \sum_{k=N+1}^{\infty} a_k^2 \le c_\mu^{\theta(\gamma+1)} (N+2)^{-2\theta(\gamma+1)} Q.$$

Hence, by setting $N \propto \epsilon^{-1/[\theta(\gamma+1)]}$, we can let $\mathcal{L}(f_{\tilde{W}}) - \mathcal{L}(f^*) \le \epsilon^2$. In this setting of $N$, by noticing $(k+1)^{-2(\theta-1)(\gamma+1)} = (k+1)^{2(1-\theta)(\gamma+1)}$ is monotonically increasing with respect to $k$, Eq. (28) gives that

$$\|\tilde{W}\|_{\mathcal{H}_{\tilde{K}}}^2 \le \sum_{k=0}^{N} c_\mu^{(\theta-1)(\gamma+1)} (k+1)^{-2(\theta-1)(\gamma+1)} a_k^2$$
$$\le c_\mu^{(\theta-1)(\gamma+1)} (N+1)^{2(1-\theta)(\gamma+1)} Q \lesssim \epsilon^{-2(1-\theta)/\theta},$$

which gives the bias term bound.

Combining the bias and variance terms, we may choose $\epsilon^*$ as the infimum of $\epsilon$ such that $\lambda_\beta \epsilon^{-\frac{2(1-\theta)}{\theta}} + (\epsilon/\lambda_\beta^{1/2})^{-\frac{2\tilde{\alpha}}{1-\tilde{\alpha}}} \le \beta \epsilon^2$. That is, we have that

$$\epsilon^{*2} \lesssim \max\left\{ \lambda_\beta^{-\tilde{\alpha}} \beta^{-(1-\tilde{\alpha})}, \left(\frac{\lambda_\beta}{\beta}\right)^{\theta}, n^{-\frac{1}{2-s}} \right\} = \max\left\{ \lambda^{-\tilde{\alpha}} \beta^{-1}, \lambda^{\theta}, n^{-\frac{1}{2-s}} \right\}.$$