[Reviews · NeurIPS 2020]

Review 1

Summary and Contributions: Summary: Analyzing deep learning optimization and connection to generalization error is a hot topic. Existing frameworks (mean field theory and Neural Tangent Kernel theory) require taking limit of infinite width to show (global) convergence. The manuscript explores a novel approach: describe deep NN training as estimating a transportation map and show convergence by analyzing a (infinite dimensional) Langevin dynamics. The authors derive excess risk bounds and even show fast rates.

Strengths: - timely and significant - wealth of methodological insights - addresses important questions

Weaknesses: Almost Nothing to declare Needs to be completed with concentration bounds (Gaussian concentration should be helpful in a first place).

Correctness: So far so good. Really checking to soundness of the paper would require learning or relearning a lot of things, months of work.

Clarity: As clear and simple a paper can be when it relies on such a variety of material. There are some unenglish sentences that could be cleaned up.

Relation to Prior Work: Yes Possible references to add: Da Prato, Giuseppe; Zabczyk, Jerzy Stochastic equations in infinite dimensions. Second edition. Encyclopedia of Mathematics and its Applications, __152__. Cambridge University Press, Cambridge, 2014. xviii+493 pp. Using Langevin dynamics to investigate training neural networks has been in the air since the days of multilayered neural networks (see Statistical mechanics of learning from examples by H. S. Seung, H. Sompolinsky, and N. Tishby Phys. Rev. A 45, 6056 – Published 1 April 1992). A. Dieuleveut's Doctoral dissertation about Stochastic Approximation in Hilbert Spaces contains relevant matrial. See http://www.cmap.polytechnique.fr/~aymeric.dieuleveut/papers/Thesis_Aymeric_Dieuleveut.pdf

Reproducibility: Yes

Additional Feedback: Stochastic Differential Equations are not the easiest topics. Especially in functional spaces. Give more hints about the spaces where solutions are supposed to live, about domain and range of operators.


Review 2

Summary and Contributions: This paper models the training process of neural networks as the estimation of a transportation map, the one taking the initial values of the weights to their values at time t (in a mean-field model of a two-layer network). The evolution of this map is studied as an infinite-dimensional Langevin dynamics and the authors show that (i) the dynamics converges to the invariant distribution under standard assumptions of ergodicity, (ii) compute the convergence rate to this invariant distribution, and (iii) compute the generalization error of solution at time k along with a bound on the excess risk. Examples are provided for binary classifier with the logistic loss and regression problems. This paper is a very terse read and a tour de force. This is a theoretical paper which could be influential.

Strengths: - The point of view of the authors, that of writing down the evolution of map as a stochastic differential equation is refreshing and quite different from the two prevalent themes (mean-field dynamics and neural tangent kernel) in the literature. - The mathematics for the excess risk bound seems novel to my understanding. - The discussion about achieving a minimax optimal "fast" rate for non-parametric regression is wonderful.

Weaknesses: - The expected excess risk bound in Theorem 2 is a lot to unpack. The Assumptions 1, 2, especially the former, should be made easier to understand. - What prevents the authors from applying this result to a non-residual deep network? It is important to discuss this.

Correctness: I have followed the main text of the paper and the theoretical results do not seem to have obvious holes.

Clarity: The current draft is a squished 8-page version of what looks like a very long and complex paper and I would encourage the authors to improve the exposition in the conference version potentially at the cost of some results.

Relation to Prior Work: This is adequately discussed.

Reproducibility: Yes

Additional Feedback:


Review 3

Summary and Contributions: This paper views the training procedure as a mapping W on the parameter space. More precisely, the mapping W itself can be viewed as one from a stochastic process of functions, induced by the training algorithm such as Langevin dynamics. Under the newly proposed framework, the optimization and generalization results are presented.

Strengths: A lifted view from the mean-field theory is very novel. Since the initial distribution can be on finite support, this framework can also apply to finite-width networks, which is an advantage compared with mean-field methods.

Weaknesses: The idea of 'lift' could cost the readers much effort to understand, especially when the idea is not clearly presented in the paper. While the new framework allows small width (compared with mean-field requiring large width to approximate the distribution), it instead needs to average over the randomness from the training algorithms. To be specific, in mean-field theory, with (undesirably) large width, the network's loss approximates the expected loss; in transportation map estimation proposed here, the expected loss of a finite-width network can be indeed bounded, but it seems not easy to guarantee the loss in one real training process is close to its expectation. Averaging over multiple training processes can address this issue, of course, but potentially it will introduces enough computation cost to make it no better than mean field.

Correctness: While I cannot go over all proof and cited results, most of the proof are correct.

Clarity: Yes.

Relation to Prior Work: Yes.

Reproducibility: Yes

Additional Feedback: Post-rebuttal: The authors' response addressed most of my concerns. I will raise my score to 6. ================================ While I find the idea of this paper very fascinating, some concerns need to be addressed. I am very willing to update my over all score given the feedback. Major concerns: 1) As mentioned in the weakness session, it seems what this paper does is moving the width requirement in the mean-field theory to the requirements of training many times. The optimization result in Eqn (7) is in expectation. Is it possible to derive a high probability bound on the training loss of one training trajectory? Instead of in expectation? If the answer is no I think the claimed improvement could look very suspicious and considered as a weakness. 2) Under your framework, the (expected) performance of the neural network function f_W depends highly on the initial distribution \rho_0. How is this reflected in your results on excessive risk? 3) In Assumption 1 (ii) , the gradient is assumed to be bounded for any w and x, would you provide an example that satisfies this? So as in Assumption 1 (iii). Minor issues: 1) The definition of ResNet in line 188-192 is very strange. How would you convince people that setting is in resemblance of any real ResNet? Also, how is the formulation applied to optimal transport? Typoes: 1) In Eqn (7) the notation for loss function L is not in \mathcal mode?


Review 4

Summary and Contributions: By formulating the neural network training as an optimal transport mapping finding problem of the parameters, this paper resolves several difficulties: (i) diverging width against sample size and (ii) curse of dimensionality. This paper mainly contains three parts: 1. It formulates neural network training as a transportation map learning of weights (parameters) and solves this problem by infinite-dimensional gradient Langevin dynamics in RKHS. 2. It shows the global convergence for the problem. 3. It also gives the generalization error bound of the estimator obtained by the proposed optimization framework.

Strengths: 1. The proposed theoretical framework covers the DNNs with finite width, which is excluded by the NTK and mean-field methods. 2. For optimization, it gives out a size-independent convergence rate. 3. For the generalization bound on the regression and the classification problem, this paper obtains the minimax and the exponential convergence, which may be hard to show in other general theoretical frameworks (without regularization).

Weaknesses: 1. The optimization part seems mainly comes from the results given by [36], and the results benefit from the regularization and the Eigen decay assumption, which may be a strong assumption since it directly helps to erase the dependence of the dimension for the bound. From this perspective, the error bound may not be so impressive. 2. The generalization conclusion also seems to profit from the regularization. With regularizer, it is always convenient to obtain certain tight generalization bounds.

Correctness: Seems correct (didn't check all the details).

Clarity: Clear writing. Minor issue: Line 312: missing reference.

Relation to Prior Work: Almost clear

Reproducibility: Yes

Additional Feedback: -------------------------------- The problem I concern is fairly discussed. I notice that the other review finds that the paper may have a gap between the analyzed SDE and the practical algorithm. Considering all the advantages and disadvantages of this paper, I insist on my original score (7).

[Author Response · NeurIPS 2020]

## Response for paper ID 8808

We thank all reviewers for their thoughtful feedback. Please find detailed responses to your comments below.

**Rev1**:

Thank you very much for carefully reading our paper and your supportive comments.

- **Unenglish sentences that could be cleaned up:** We will make the manuscript proofread by a professional English editing service, which we believe resolves grammatical issues.

**Rev2**:

Thank you very much for carefully reading our paper and your supportive comments.

- **Theorem 2 is a lot to unpack. Assumptions 1, 2 should be made easier:** The analysis tends to be complicated because the infinite dimensional Langevin dynamics requires a few involved conditions such as smoothness on the objective functions. We are doing our best to make the results presented as intuitively as possible. We will add more digested expositions to the assumptions and convergence rate analysis.

- **Application to non-residual deep network. It is important to discuss this:** Thank you very much for pointing our an important point. Indeed, our approach *can* be applied to non-residual deep network. The reason why we presented ResNet is just due to space limitation and the fact that ResNet has a continuous-depth representation and such a continuous depth representation is also an interesting application of our analysis. Since we had this application in our mind, we have presented only ResNet in the main text. We will add some more comments about this point in the final version.

- **I would encourage the authors to improve the exposition:** Thank you for your suggestive comment. We tried to keep technical details as much as possible so that there does not occur confusion and misunderstanding. However, as you pointed out, we would like to use more spaces for intuitive expositions and move some details to the appendix.

**Rev3**:

- **Is it possible to derive a high probability bound on the training loss of one training trajectory?:** Yes, the most direct way is to apply the Markov's inequality. Moreover, the expectation with respect to the training sample observation is derived from a exponential tail probability bound and thus we can derive a high probability bound with respect to sample observation. As for the training trajectory, the mixing time of the dynamics is fast (exponential with respect to the iteration) and thus as the iteration number increases, the probability in which the trajectory does not contain a "nice" solution satisfying the risk bound decreases exponentially to 0. Since the high probability bound makes the statements complicated (the technical contents are already a bit involved), we have shown the expectation bound for simplicity. We will add more comments on the high probability bound in the final version.

- **How is $\rho_0$ reflected in your results on excessive risk?:** As the algorithm progresses, the solution "forgets" the initial solution exponentially fast. In that sense, it does not affect so much on the excess risk. On the other hand, the concentration function is characterized by the relative location between the optimal solution and $\mathcal{H}_K$, the geometry of $L_2(\rho_0)$ indirectly affects the excess risk through the shape of $\mathcal{H}_K$. However, it is highly problem dependent.

- **Assumption 1 (ii), (iii) :** The two layer neural network model presented in the excess risk bound satisfies these conditions (Eq.(8) with bounded input $\|x\| \leq D$ (a.s.) and smooth loss). More specifically, under the setting of Theorem 2, Assumption 1 is satisfied.

- **ResNet in line 188-192 is strange:** We would like to remark that this is a standard definition where the residual blocks are two layer neural networks. Each layer $\ell$ receives an output from the previous layer as $x_\ell$ and it outputs $x_\ell + g_\ell(x_\ell) = (I + g_\ell(\cdot))x_\ell$ to the next layer where $g_\ell$ is a two layer neural network given as $g_\ell(x) = \int a_{w,\ell}\sigma(W(w,\ell)^\top x)\mathrm{d}\rho_0(w)$. This formulation is standard in theoretical analyses of ResNet (e.g., [1, 2]).

**Rev4**:

- **Eigen decay may be a strong assumption since it directly helps to erase the dependence of the dimension for the bound. With regularizer, it is always convenient to obtain certain tight generalization bounds:** Indeed, as you pointed out, regularization is the most essential ingredient to obtain a width free generalization bound. Conversely, we can not expect nice generalization without any regularization. Although a global optimal solution for non-convex loss does not directly indicate good generalization, our analysis connects generalization and algorithmic convergence, which we believe is an interesting point. In a real deep learning, we consider that such a regularization is imposed through several explicit/implicit regularizations.

## References

[1] Y. Lu, C. Ma, Y. Lu, J. Lu, and L. Ying. A mean-field analysis of deep ResNet and beyond: Towards provable optimization via overparameterization from depth. In *Proceedings of ICML2020*, pages 137–147, 2020.

[2] E. Weinan, J. Han, and Q. Li. A mean-field optimal control formulation of deep learning. *Research in the Mathematical Sciences*, 6(1):10, 2019.


[Meta-Review · NeurIPS 2020]

Four knowledgeable referees consider this a timely and significant contribution with several methodological insights and novelties, which could be influential. A few concerns are the dense presentation and possible difficulties with interpretation and the role of regularization. I would like to encourage the authors to make an effort to address these points and try to make the contents more accessible in the final manuscript. I am recommending a spotlight accept.